# NeuralGF: Unsupervised Point Normal Estimation by Learning Neural Gradient Function

**Qing Li**[1]    **Huifang Feng**[2]    **Kanle Shi**[3]    **Yue Gao**[1]    **Yi Fang**[4]
**Yu-Shen Liu**[1][*]    **Zhizhong Han**[5]

[1]School of Software, Tsinghua University, Beijing, China
[2]School of Informatics, Xiamen University, Xiamen, China
[3]Kuaishou Technology, Beijing, China
[4]Center for Artificial Intelligence and Robotics, New York University Abu Dhabi, Abu Dhabi, UAE
[5]Department of Computer Science, Wayne State University, Detroit, USA
{leoqli, gaoyue, liuyushen}@tsinghua.edu.cn    fenghuifang@stu.xmu.edu.cn
yfang@nyu.edu    h312h@wayne.edu

## Abstract

Normal estimation for 3D point clouds is a fundamental task in 3D geometry processing. The state-of-the-art methods rely on priors of fitting local surfaces learned from normal supervision. However, normal supervision in benchmarks comes from synthetic shapes and is usually not available from real scans, thereby limiting the learned priors of these methods. In addition, normal orientation consistency across shapes remains difficult to achieve without a separate post-processing procedure. To resolve these issues, we propose a novel method for estimating oriented normals directly from point clouds without using ground truth normals as supervision. We achieve this by introducing a new paradigm for learning neural gradient functions, which encourages the neural network to fit the input point clouds and yield unit-norm gradients at the points. Specifically, we introduce loss functions to facilitate query points to iteratively reach the moving targets and aggregate onto the approximated surface, thereby learning a global surface representation of the data. Meanwhile, we incorporate gradients into the surface approximation to measure the minimum signed deviation of queries, resulting in a consistent gradient field associated with the surface. These techniques lead to our deep unsupervised oriented normal estimator that is robust to noise, outliers and density variations. Our excellent results on widely used benchmarks demonstrate that our method can learn more accurate normals for both unoriented and oriented normal estimation tasks than the latest methods. The source code and pre-trained model are available at https://github.com/LeoQLi/NeuralGF.

## 1   Introduction

Normal vectors are one of the local descriptors of point clouds and can offer additional geometric information for many downstream applications, such as denoising [70, 45, 44, 43], segmentation [63–65] and registration [62, 68]. Moreover, some computer vision tasks require the normals to have consistent orientations, *i.e.*, oriented normals, such as graphics rendering [10, 21, 61] and surface reconstruction [26, 31–33]. Therefore, point cloud normal estimation has been an important research topic for a long time. However, progress in this area has plateaued for some time, until the recent introduction of deep learning. Currently, a growing body of work shows that performance improvements can be achieved using data-driven approaches [24, 8, 40, 39], and these learning-based methods often

---

[*]Corresponding author

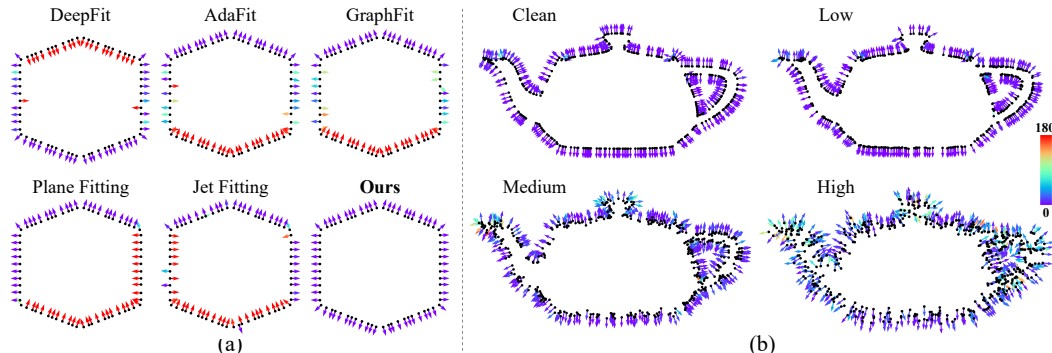

Figure 1: (a) Existing surface fitting-based normal estimation methods can only produce unoriented normals (arrows) with inconsistent orientations. (b) Our method can estimate oriented normals (arrows) from unevenly sampled point clouds (black dots) with varying noise levels. The color of the arrow indicates the angle error compared to the ground truth.

give better results than traditional data-independent methods. However, they suffer from shortcomings in terms of robustness to various data and computational efficiency [36]. Besides, they usually rely on a large amount of labeled training data and network parameters to learn high-dimensional features. More importantly, the presence of varying noise levels, uneven sampling densities, and various complex geometries poses challenges for accurately estimating oriented normals from point clouds.

Most existing point cloud normal estimation methods aim to locally fit geometric surfaces and compute normals from the fitted surfaces. Recent surface fitting-based methods, such as DeepFit [8], AdaFit [86] and GraphFit [38], generalize the $n$-jet surface model [15] to learning-based frameworks that predict pointwise weights using deep neural networks. They obtain surface normals by solving the weighted least squares polynomial fitting problem. However, as shown in Fig. 1(a), their methods can not guarantee consistent orientations very well, since $n$-jet only fits a local surface, which is ambiguous for determining orientation. Furthermore, their methods need to construct local patches based on each query point for surface fitting. This results in the algorithm needing to traverse all patches in the point cloud, which is time-consuming. Most importantly, they rely on the priors learned from normal supervision to fit local surfaces, while such supervision is not always available.

Inspired by neural implicit representations [6, 7, 46, 49, 16], we introduce neural gradient functions (NeuralGF) to approximate the global surface representation and constrain the gradient during the approximation. Our method includes two basic principles: (1) We formulate the neural gradient learning as iteratively learning an implicit surface representation directly from the entire point cloud for global surface fitting. (2) We constrain the derived surface gradient vector to accurately describe the local distribution and make it consistent across iterations. Accordingly, we implement these two principles using the designed loss functions, resulting in a model that can provide accurate normals with consistent orientations from the entire point cloud (see Fig. 1(b)) through a single forward pass, and does not require ground truth labels or post-processing. Our method improves the state-of-the-art results while using much fewer parameters. To summarize, our main contributions include:

- Introducing a new paradigm of learning neural gradient functions, building on implicit geometric representations, for computing consistent gradients directly from raw data without ground truth.
- Implementing the paradigm with the losses designed to train neural networks to fit global surfaces and estimate oriented normals with consistent orientations in an end-to-end manner.
- Reporting the state-of-the-art performance for both unoriented and oriented normal estimation on point clouds with noise, density variations, and complex geometries.

## 2 Related Work

**Unoriented Normal Estimation**. Point cloud normal estimation has been extensively studied over the past decades and widely used in many downstream applications. In general, most of these studies focus on estimating unoriented normals, *i.e.*, finding more accurate perpendiculars of local surfaces. The Principle Component Analysis (PCA) [26] is the most commonly used algorithm that has been widely applied to various geometric processing tasks. Later, PCA variants [1, 59, 54, 35, 27], Voronoi

diagrams [3, 18, 2, 51] and Hough transform [11] have been proposed to improve the performance by preserving the shape details or sharp features. Moreover, some methods [37, 15, 23, 56, 4] are designed to fit complex surfaces by containing more neighboring points, such as moving least squares [37], truncated Taylor expansion ($n$-jet) [15] and spherical surface fitting [23]. These methods usually suffer from the parameters tuning, noise and outliers. To achieve more robust capability, they usually choose a larger neighborhood size but over-smooth the details. In recent years, deep learning-based normal estimation methods have been proposed and achieve great success. The regression-based methods predict normals from structured data or raw point clouds. Specifically, some methods [12, 66, 43] transform the 3D points into structured data, such as 2D grid representations. Inspired by PointNet [63], other methods [24, 83, 25, 9, 81, 82, 40, 41, 19, 39] focus on the direct regression for unstructured point clouds. In contrast to regressing the normal vectors directly, the surface fitting-based methods aim to train a neural network to predict pointwise weights, then the normals are solved through weighted plane fitting [36, 14] or weighted polynomial surface fitting [8, 86, 84, 80, 38] on local neighborhoods. Although these deep learning-based methods perform better than traditional data-independent methods, they require ground truth normals as supervision. In this work, we propose to use networks to estimate normals in an unsupervised manner.

**Consistent Normal Orientation**. Estimating normals with consistent orientations is also important and has been widely studied. The most popular and simplest way for normal orientation is based on local propagation, where the orientation of a seed point is diffused to the adjacent points by a Minimum Spanning Tree (MST). Most existing normal orientation approaches are based on this strategy, such as the pioneering work [26] and its improved methods [34, 69, 72, 67, 76, 29]. More recent work [53] integrates a neural network to learn oriented normals within a single local patch and introduces a dipole propagation strategy across the partitioned patches to achieve global consistency. These propagation-based approaches usually assume smoothness and suffer from local error propagation and the choice of neighborhood size. Since local consistency is usually not enough to achieve robust orientations across different inputs, some other methods determine the normal orientation by applying various volumetric representation techniques, such as signed distance functions [50, 55], variational formulations [71, 2, 28], visibility [30, 17], active contours [75], isovalue constraints [74] and winding-number field [77]. In addition to the above approaches, the most recent researches [24, 25, 73, 39] aim to learn oriented normals directly from point clouds in a data-driven manner. Similar to the unoriented normal estimation task, these deep learning-based methods perform better than traditional methods, but they require expensive ground truth as supervision. In contrast, our method achieves better performance in an unsupervised manner.

**Neural Implicit Representation**. Recent works [52, 57] propose to learn neural implicit representation from point clouds. Later, one class of methods [6, 7, 46, 60, 85, 47, 48] focuses on training neural networks to overfit a single point cloud by introducing newly designed priors. SAL [6] uses unsigned regression to find signed local minima, thereby producing useful implicit representations directly from raw 3D data. IGR [22] proposes an implicit geometric regularization to learn smooth zero level set surfaces. Neural-Pull [46] uses the signed distance to move a point along the gradient for finding its nearest path to the surface. SAP [60] proposes a differentiable Poisson solver to represent the surface as an oriented point cloud at the zero level set of an indicator function. These methods focus on accurately locating the position of the zero iso-surface of an implicit function to extract the shape surface. However, they ignore the constraints on the gradient of the function during optimizing the neural network. We know that the gradient determines the direction of function convergence, and the gradient of the iso-surface can be used as the normal of the extracted surface. If the gradient can be guided reasonably, the convergence process can be more robust and efficient, avoiding local extremum caused by noise or outliers. Motivated by this, we try to incorporate neural gradients into implicit function learning to achieve accurate oriented normals.

## 3 Method

**Preliminary**. Neural implicit representations are widely used in representing 3D geometry and have achieved promising performance in surface reconstruction. Benefiting from the adaptability and approximation capabilities of deep neural networks [5], recent approaches [57, 22, 46, 48] use them to learn signed distance fields as a mapping from 3D coordinates $\boldsymbol{x}$ to signed distances. These approaches represent surfaces as zero level sets of implicit functions $f$, *i.e.*, $\mathcal{S} = \left\{ \boldsymbol{x} \in \mathbb{R}^3 \mid f(\boldsymbol{x}; \boldsymbol{\theta}) = 0 \right\}$, where $f \colon \mathbb{R}^3 \to \mathbb{R}$ is a network with parameter $\boldsymbol{\theta}$, which is usually implemented as a multi-layer perceptron

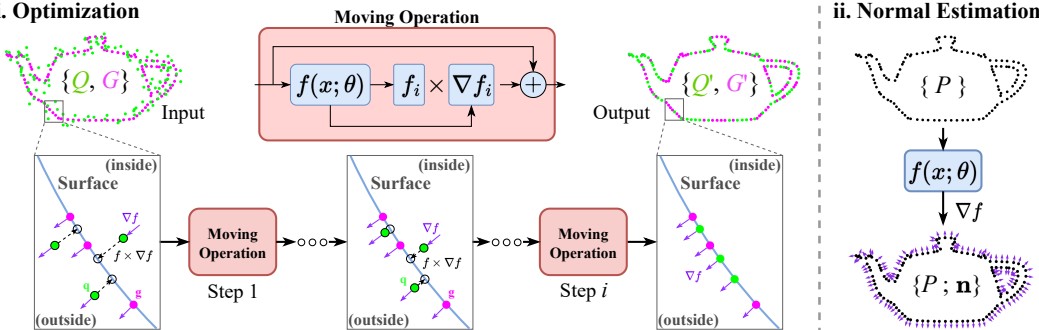

Figure 2: Overview of our method. The optimization is formulated as an iterative moving operation of the positions of point set $\{Q, G\}$, which is sampled from the raw data $P$, while the normal estimation is a single forward differentiation of the learned function $f(x; \theta)$ using $P$ as input. We move the input set $\{Q, G\}$ through multiple steps using signed distance $f_i$ and gradient $\nabla f_i$ learned from $f(x; \theta)$ to obtain the output set $\{Q', G'\}$. The gradient is initialized to point outside the surface.

(MLP). If the function $f$ is continuous and differentiable, the normal of a point $p$ on the surface can be represented as $\mathbf{n}_p = \nabla f(p) / \|\nabla f(p)\|$, where $\|\cdot\|$ denotes the euclidean $L^2$-norm and $\nabla f(p)$ indicates the gradient at $p$, which can be obtained in the differentiation process of $f$. Specifically, there are some important properties of the gradient: (1) The maximum rate of change of the function $f$ is defined by the magnitude of the gradient $\|\nabla f\|$ and given by the direction of $\nabla f$. (2) The gradient vector $\nabla f(p)$ is perpendicular to the level surface $f(p) = 0$.

**Overview**. We aim to train a neural network $f$ with parameter $\boldsymbol{\theta}$ from input points $\boldsymbol{x}$ to obtain the gradient $\nabla f$ during the optimization of $f$. The optimization encourages $f$ to fit $\boldsymbol{x}$ and the gradient $\nabla f$ to approach the normal of the implicit surface. We sample a query point set $Q = \{q_j\}_{j=1}^{N_Q}$ and a surface point set $G = \{g_j\}_{j=1}^{N_G}$ from raw point cloud $P$. Specifically, $Q$ is sampled through some probability distribution $\mathcal{D}$ from $P$ [13, 46]. To explicitly indicate the underlying geometry, we sample the point set $G \subset P$ that may be distorted by noise, and we verify that it can facilitate the learning of $f$ (see Sec. 4.2). As shown in Fig. 2, our optimization pipeline is formulated as an iterative moving operation of the union of points $Q$ and $G$, more specifically, with $\mathcal{I}$ steps of position movement. During normal estimation, the entire point cloud $P$ is fed into the network to derive the gradient embedded in the learned function. To learn the neural gradient function based on the implicit surface representation of $f$, we consider a loss of the form

$$\mathcal{L}(\boldsymbol{\theta}) = \mathbb{E}_{\boldsymbol{x} \sim \mathcal{D}} \Big( \mathcal{T}\big(V(\boldsymbol{x}; \boldsymbol{\theta}), \, \mathcal{V}(\boldsymbol{x})\big) + \lambda \, \mathcal{T}\big(F(\boldsymbol{x}; \boldsymbol{\theta}), \, \mathcal{F}(\boldsymbol{x})\big) \Big), \tag{1}$$

where $\mathcal{T} : \mathbb{R} \times \mathbb{R} \to \mathbb{R}$ is a differentiable similarity function, and we adopt the standard $L^2$ (Euclidean) distance. $V(\boldsymbol{x}; \boldsymbol{\theta})$ and $F(\boldsymbol{x}; \boldsymbol{\theta})$ represent the approximated gradient vector and global surface representation of the underlying geometry $P$, respectively. $\mathcal{V}(\boldsymbol{x})$ and $\mathcal{F}(\boldsymbol{x})$ denote the corresponding measures, respectively. We will introduce them in the following two sections. $\lambda > 0$ is a weighting factor. The expectation $\mathbb{E}$ is made for certain probability distribution $\boldsymbol{x} \sim \mathcal{D}$ in 3D space.

## 3.1 Learning Global Surface

The $n$-jet surface model represents a polynomial function that maps points $(x, y) \in \mathbb{R}^2$ to their height $z$ that is not in the tangent space over a local surface [8, 86, 38]. In contrast, we aim to learn an implicit global surface representation of the point cloud. The gradient indicates the direction in 3D space in which the signed distance from the surface increases the fastest, so moving a point along the gradient will find its nearest path to the surface. According to this property, we adopt a moving operation introduced in [46] to project a query point $q$ to $q'$, and $q' = q - f(q; \boldsymbol{\theta}) \cdot \mathbf{n}_q$, $\mathbf{n}_q = \nabla f(q; \boldsymbol{\theta}) / \|\nabla f(q; \boldsymbol{\theta})\|$. We also use $\nabla f$ to denote the normalized gradient for simplification. For the point $q \in Q$, we expect the function $f$ could provide correct signed distance $f(q; \boldsymbol{\theta})$ and gradient $\nabla f(q; \boldsymbol{\theta})$ that can be used to move $q$ to the nearest point $\hat{q}$ over the surface, and we minimize the error $\|q' - \hat{q}\|$. In this way, the function $f$ can be used as an implicit representation of the underlying surface and the zero level set of $f$ will be a valid manifold describing the point cloud. In

this work, the moving operation for input point set $\{Q, G\}$ is formulated as

$$\mathcal{L}_d(\boldsymbol{\theta}) = \mathbb{E}_{\boldsymbol{x} \sim \mathcal{D}} \mathcal{T}\big(F(\boldsymbol{x}; \boldsymbol{\theta}), \mathcal{F}(\boldsymbol{x})\big) = \left\| \{Q', G'\} - \{\hat{Q}, \hat{G}\} \right\|, \quad \{Q', G'\} = \{Q, G\} - \sum_{i=1}^{\mathcal{I}} f_i \cdot \nabla f_i, \quad (2)$$

where $\{Q', G'\}$ is the position of point set $\{Q, G\}$ after moving in multiple steps $\mathcal{I}$ (see Fig. 2), and $\{\hat{Q}, \hat{G}\}$ is the target position. $f_i$ represents the signed distance of points during the $i$-th position movement step. In practice, we find the nearest point of $Q$ and $G$ from the raw point cloud $P$ as the target position. Different from previous works [46–48], we introduce a multi-step movement strategy to optimize the point set $\{Q', G'\}$ to cover the surface instead of single-step movement. The insight of our design is that it is always difficult to directly reach the target in one step if some points are distributed far from the surface or distorted by noise. At this time, the optimization of the moving operation may go through many twists and turns to converge.

We optimize $\mathcal{L}_d(\boldsymbol{\theta})$ in an end-to-end manner based on the moved point set $\{Q', G'\}$. For the points $G \subset P$ in the first step, we expect them to be on the underlying surface. For the points $\{Q', G'\}$ moved after the first step, we expect them to be as close to the underlying surface as possible. Hence, we leverage a regression loss term on the predicted distances $f_i^Q$ and $f_i^G$ of point set $\{Q, G\}$, i.e.,

$$\mathcal{L}_{reg}(\boldsymbol{\theta}) = \left(f_1^G\right)^2 + \sum_{i=2}^{\mathcal{I}} (f_i)^2, \quad \text{and} \quad f_i = \{f_i^Q, f_i^G\}. \quad (3)$$

## 3.2 Learning Consistent Gradient

A standard way for estimating unoriented point normals is to fit a plane to the local neighborhood $\mathcal{N}(p, P)$ of each point $p$ [37]. Given a neighborhood size $K$, we let $\mathcal{V}(p) \in \mathbb{R}^{K \times 3}$ denote the set of centered coordinates of points in that neighborhood. Then the plane fitting is described as finding the least squares solution of

$$\mathbf{n}_p^* = \arg \min_{\mathbf{n}: \|\mathbf{n}\|=1} \sum_{k=1}^{K} \|\mathbf{n} \cdot \mathcal{V}(p)_k\|^2, \quad \text{and} \quad \mathcal{V}(p)_k = p_k - \frac{1}{K} \sum_{l=1}^{K} p_l, \quad p_k, p_l \in \mathcal{N}(p, P). \quad (4)$$

However, the inner product in the formula allows the distribution of normals on both sides of the surface (positive or negative orientations) to satisfy the optimization. Therefore, this approach cannot determine the normal orientation with respect to the surface, and a consistent orientation needs to be solved by post-processing, such as MST-based orientation propagation. In this work, we directly estimate oriented normals by incorporating neural gradients into implicit function learning [6, 7],

$$\mathcal{L}_v(\boldsymbol{\theta}) = \mathbb{E}_{\boldsymbol{x} \sim \mathcal{D}} \mathcal{T}\big(V(\boldsymbol{x}; \boldsymbol{\theta}), \mathcal{V}(\boldsymbol{x})\big) = \|f(\boldsymbol{x}; \boldsymbol{\theta}) \cdot \mathbf{n} - \mathcal{V}(\boldsymbol{x})\|, \quad (5)$$

where $\mathbf{n}$ is the unit gradient (normal) of each point on the surface. The above formula aims to measure the minimum deviation of each point from the underlying surface. Specifically, the gradient is perpendicular to the surface and the distance from the surface increases the fastest along the direction of the gradient. Therefore, we measure the deviation from $P$ to $Q$ by finding the optimal gradients. Generally, raw point clouds may contain noise and outliers that can severely degrade the accuracy of the surface approximation. To reduce the impact of inaccuracy in $P$, we introduce multi-scale neighborhood size $\{K_s\}_{s=1}^{N_K}$ for $\mathcal{V}(x)$ based on a statistical manner, that is

$$\mathcal{V}_s(q) = q - \frac{1}{K_s} \sum_{k=1}^{K_s} q_k, \quad q_k \in \mathcal{N}_{K_s}(q, P), \quad s = 1, \cdots, N_K, \quad (6)$$

where $\mathcal{N}_{K_s}(q, P)$ denotes the $K_s$ nearest points of $q \in Q$ in $P$. $\mathcal{V}(q)$ is built as a vector from the averaged position $\bar{\boldsymbol{x}} = \sum_{k=1}^{K_s} \mathcal{N}_{K_s}(q, P)/K_s$ on surface to query $q$. Then, the objective of Eq. (5) is turned into the aggregation of the error at each neighborhood size scale, i.e.,

$$\mathcal{L}_v(\boldsymbol{\theta}) = \mathbb{E}_{\boldsymbol{x} \sim \mathcal{D}} \mathcal{T}\big(V(\boldsymbol{x}; \boldsymbol{\theta}), \mathcal{V}(\boldsymbol{x})\big) = \sum_{s=1}^{N_K} \left\| f_i^Q \cdot \nabla f_i^Q - \mathcal{V}_s(Q) \right\|, \quad (7)$$

where $i = 1, \cdots, \mathcal{I}$ is the number of steps. In practice, we only use $f_1$ of the first step and its $\nabla f_1$ to measure the deviation, since we want the optimization to converge as fast as possible, which also implies that the position $Q'$ after the first move step in Eq. (2) is as close to the surface as possible.

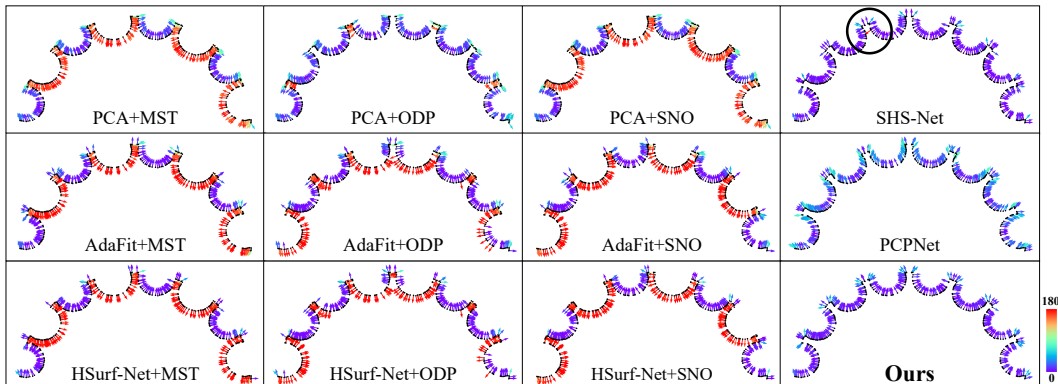

Figure 3: Visual comparison of oriented normals. We only show a partial cross-section of a closed shape for better visualization. The color of the arrow indicates RMSE, where purple means the same orientation as the ground truth and red means the opposite.

Table 1: RMSE of oriented normals on datasets PCPNet and FamousShape. Our method achieves SOTA performance even when compared to supervised baselines. ∗ means the code is uncompleted.

| Category | PCPNet Dataset | | | | | | | FamousShape Dataset | | | | | | |
|---|---|---|---|---|---|---|---|---|---|---|---|---|---|---|
| | Noise | | | | Density | | Average | Noise | | | | Density | | Average |
| | None | 0.12% | 0.6% | 1.2% | Stripe | Gradient | | None | 0.12% | 0.6% | 1.2% | Stripe | Gradient | |
| PCA [26]+MST [26] | 19.05 | 30.20 | 31.76 | 39.64 | 27.11 | 23.38 | 28.52 | 35.88 | 41.67 | 38.09 | 60.16 | 31.69 | 35.40 | 40.48 |
| PCA [26]+SNO [67] | 18.55 | 21.61 | 30.94 | 39.54 | 23.00 | 25.46 | 26.52 | 32.25 | 39.39 | 41.80 | 61.91 | 36.69 | 35.82 | 41.31 |
| PCA [26]+ODP [53] | 28.96 | 25.86 | 34.91 | 51.52 | 28.70 | 23.00 | 32.16 | 30.47 | 31.29 | 41.65 | 84.00 | 39.41 | 30.72 | 42.92 |
| LRR [79]+MST [26] | 43.48 | 47.58 | 38.58 | 44.08 | 48.45 | 46.77 | 44.82 | 56.24 | 57.38 | 45.73 | 64.63 | 66.35 | 56.65 | 57.83 |
| LRR [79]+SNO [67] | 44.87 | 43.45 | 33.46 | 45.40 | 46.96 | 37.73 | 41.98 | 59.78 | 60.18 | 45.02 | 71.37 | 62.78 | 59.90 | 59.84 |
| LRR [79]+ODP [53] | 28.65 | 25.83 | 36.11 | 53.89 | 26.41 | 23.72 | 32.44 | 39.97 | 42.17 | 48.29 | 88.68 | 44.92 | 47.56 | 51.93 |
| AdaFit [86]+MST [26] | 27.67 | 43.69 | 48.83 | 54.39 | 36.18 | 40.46 | 41.87 | 43.12 | 39.33 | 62.28 | 60.27 | 45.57 | 42.00 | 48.76 |
| AdaFit [86]+SNO [67] | 26.41 | 24.17 | 40.31 | 48.76 | 27.74 | 31.56 | 33.16 | 27.55 | 37.60 | 69.56 | 62.77 | 27.86 | 29.19 | 42.42 |
| AdaFit [86]+ODP [53] | 26.37 | 24.86 | 35.44 | 51.88 | 26.45 | 20.57 | 30.93 | 41.75 | 39.19 | 44.31 | 72.91 | 45.09 | 42.37 | 47.60 |
| HSurf-Net [40]+MST [26] | 29.82 | 44.49 | 50.47 | 55.47 | 40.54 | 43.15 | 43.99 | 54.02 | 42.67 | 68.37 | 65.91 | 52.52 | 53.96 | 56.24 |
| HSurf-Net [40]+SNO [67] | 30.34 | 32.34 | 44.08 | 51.71 | 33.46 | 40.49 | 38.74 | 41.62 | 41.06 | 67.41 | 62.04 | 45.59 | 43.83 | 50.26 |
| HSurf-Net [40]+ODP [53] | 26.91 | 24.85 | 35.87 | 51.75 | 26.91 | 20.16 | 31.07 | 43.77 | 43.74 | 46.91 | 72.70 | 45.09 | 43.98 | 49.37 |
| PCPNet [24] | 33.34 | 34.22 | 40.54 | 44.46 | 37.95 | 35.44 | 37.66 | 40.51 | 41.09 | 46.67 | 54.36 | 40.54 | 44.26 | 44.57 |
| DPGO∗ [73] | 23.79 | 25.19 | 35.66 | 43.89 | 28.99 | 29.33 | 31.14 | - | - | - | - | - | - | - |
| SHS-Net [39] | **10.28** | **13.23** | 25.40 | 35.51 | 16.40 | 17.92 | 19.79 | 21.63 | 25.96 | 41.14 | 52.67 | 26.39 | 28.97 | 32.79 |
| Ours | 10.60 | 18.30 | **24.76** | **33.45** | **12.27** | **12.85** | **18.70** | **16.57** | **19.28** | **36.22** | **50.27** | **17.23** | **17.38** | **26.16** |

In addition, we pursue better gradient consistency across iterations, *i.e.*, make the gradient directions in each move step parallel. To achieve this, we align the gradient $\nabla f_i$ after the first step with the gradient $\nabla f_1$ of the first step. Meanwhile, we also consider the confidence of each point with respect to the underlying surface. Thus, we introduce confidence-weighted cosine distance to evaluate the gradient consistency at points $\{Q, G\}$, with the form of

$$\mathcal{L}_{con}(\boldsymbol{\theta}) = \sum_{i=2}^{\mathcal{I}} w \cdot \left(1 - \langle \nabla f_1, \nabla f_i \rangle\right), \text{ and } \nabla f_i = \left\{\nabla f_i^Q, \nabla f_i^G\right\}, \tag{8}$$

where $\langle \cdot \rangle$ denotes cosine distance of vectors. $w = \exp(-\rho \cdot |f_1|)$ is an adaptive weight that indicates the significance of each input point based on the predicted distance, making the model pay more attention to points near the surface. In summary, our method combines an implicit representation and a gradient approximation with respect to the underlying geometry of $P$. We not only consider the gradient consistency at $q$ during iterations, but also unify the adjacent regions of $q$ to compensate for the inaccuracy of $P$. This is important because the input point cloud may be noisy and some points are not on the underlying surface.

**Loss Function**. We optimize global surface approximation and consistent gradient learning in an end-to-end manner by adjusting the parameter $\boldsymbol{\theta}$ of $f$ using the following loss function

$$\mathcal{L}(\boldsymbol{\theta}) = \mathcal{L}_v(\boldsymbol{\theta}) + \lambda_1 \mathcal{L}_{con}(\boldsymbol{\theta}) + \lambda_2 \mathcal{L}_d(\boldsymbol{\theta}) + \lambda_3 \mathcal{L}_{reg}(\boldsymbol{\theta}), \tag{9}$$

where $\lambda_1 = 0.01$, $\lambda_2 = 0.1$ and $\lambda_3 = 10$ are balance weights and are fixed across all experiments.

**Neural Gradient Computation**. The above losses require incorporating the gradient $\nabla f(\boldsymbol{x}; \boldsymbol{\theta})$ in a differentiable manner. As in [22, 7], we construct $\nabla f(\boldsymbol{x}; \boldsymbol{\theta})$ as a neural network in conjunction with

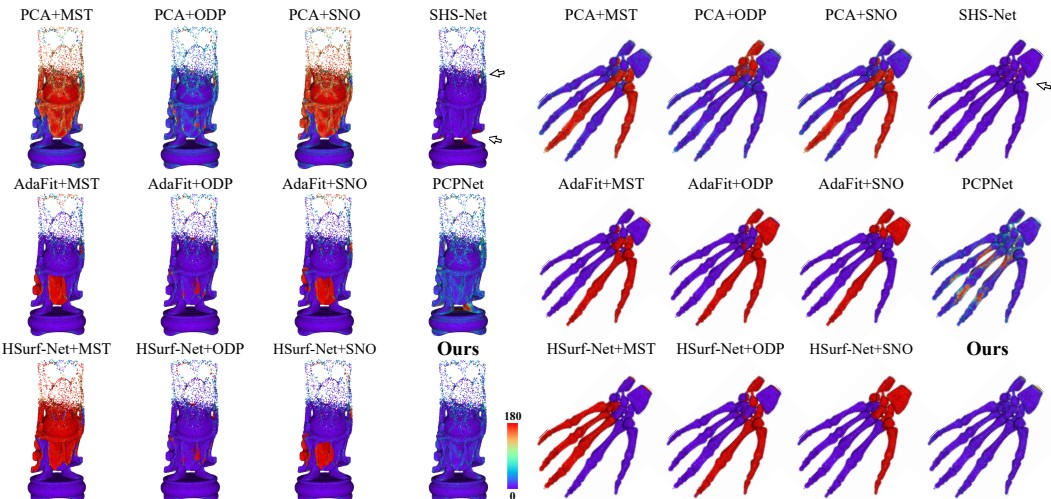

Figure 4: Visual comparison of oriented normals on point clouds with density variations and complex geometries. Point color indicates normal RMSE and red (180°) indicates reversed orientation.

Table 2: RMSE of unoriented normal on datasets PCPNet and FamousShape. Our method achieves SOTA performance compared to unsupervised baseline methods.

| Category | PCPNet Dataset | | | | | | | FamousShape Dataset | | | | | | |
|---|---|---|---|---|---|---|---|---|---|---|---|---|---|---|
| | None | Noise 0.12% | 0.6% | 1.2% | Density Stripe | Gradient | Average | None | Noise 0.12% | 0.6% | 1.2% | Density Stripe | Gradient | Average |
| **Supervised** | | | | | | | | | | | | | | |
| PCPNet [24] | 9.64 | 11.51 | 18.27 | 22.84 | 11.73 | 13.46 | 14.58 | 18.47 | 21.07 | 32.60 | 39.93 | 18.14 | 19.50 | 24.95 |
| Nesti-Net [9] | 7.06 | 10.24 | 17.77 | 22.31 | 8.64 | 8.95 | 12.49 | 11.60 | 16.80 | 31.61 | 39.22 | 12.33 | 11.77 | 20.55 |
| Lenssen *et al.* [36] | 6.72 | 9.95 | 17.18 | 21.96 | 7.73 | 7.51 | 11.84 | 11.62 | 16.97 | 30.62 | 39.43 | 11.21 | 10.76 | 20.10 |
| DeepFit [8] | 6.51 | 9.21 | 16.73 | 23.12 | 7.92 | 7.31 | 11.80 | 11.21 | 16.39 | 29.84 | 39.95 | 11.84 | 10.54 | 19.96 |
| Zhang *et al.* [80] | 5.65 | 9.19 | 16.78 | 22.93 | 6.68 | 6.29 | 11.25 | 9.83 | 16.13 | 29.81 | 39.81 | 9.72 | 9.19 | 19.08 |
| AdaFit [86] | 5.19 | 9.05 | 16.45 | 21.94 | 6.01 | 5.90 | 10.76 | 9.09 | 15.78 | 29.78 | 38.74 | 8.52 | 8.57 | 18.41 |
| GraphFit [38] | 5.21 | 8.96 | 16.12 | 21.71 | 6.30 | 5.86 | 10.69 | 8.91 | 15.73 | 29.37 | 38.67 | 9.10 | 8.62 | 18.40 |
| NeAF [41] | 4.20 | 9.25 | 16.35 | 21.74 | 4.89 | 4.88 | 10.22 | 7.67 | 15.67 | 29.75 | 38.76 | 7.22 | 7.47 | 17.76 |
| HSurf-Net [40] | 4.17 | 8.78 | 16.25 | 21.61 | 4.98 | 4.86 | 10.11 | 7.59 | 15.64 | 29.43 | 38.54 | 7.63 | 7.40 | 17.70 |
| Du *et al.* [19] | 3.85 | 8.67 | 16.11 | 21.75 | 4.78 | 4.63 | 9.96 | 6.92 | 15.05 | 29.49 | 38.73 | 7.19 | 6.92 | 17.38 |
| SHS-Net [39] | 3.95 | 8.55 | 16.13 | 21.53 | 4.91 | 4.67 | 9.96 | 7.41 | 15.34 | 29.33 | 38.56 | 7.74 | 7.28 | 17.61 |
| **Unsupervised** | | | | | | | | | | | | | | |
| Boulch *et al.* [11] | 11.80 | 11.68 | 22.42 | 35.15 | 13.71 | 12.38 | 17.86 | 19.00 | 19.60 | 36.71 | 50.41 | 20.20 | 17.84 | 27.29 |
| PCV [78] | 12.50 | 13.99 | 18.90 | 28.51 | 13.08 | 13.59 | 16.76 | 21.82 | 22.20 | 31.61 | 46.13 | 20.49 | 19.88 | 27.02 |
| Jet [15] | 12.35 | 12.84 | **18.33** | 27.68 | 13.39 | 13.13 | 16.29 | 20.11 | 20.57 | 31.34 | 45.19 | 18.82 | 18.69 | 25.79 |
| PCA [26] | 12.29 | 12.87 | 18.38 | 27.52 | 13.66 | 12.81 | 16.25 | 19.90 | 20.60 | 31.33 | 45.00 | 19.84 | 18.54 | 25.87 |
| LRR [79] | 9.63 | 11.31 | 20.53 | 32.53 | 10.42 | 10.02 | 15.74 | 17.68 | 19.32 | 33.89 | 49.84 | 16.73 | 16.33 | 25.63 |
| Ours | **7.89** | **9.85** | 18.62 | **24.89** | **9.21** | **9.29** | **13.29** | **13.74** | **16.51** | **31.05** | **40.68** | **13.95** | **13.17** | **21.52** |

$f(\boldsymbol{x}; \boldsymbol{\theta})$. Specifically, each layer of the network has the form $\boldsymbol{y}^{\ell+1} = \phi(\boldsymbol{W}\boldsymbol{y}^{\ell} + \boldsymbol{b})$, where $\boldsymbol{\theta} = (\boldsymbol{W}, \boldsymbol{b})$ is the learnable parameter of $\ell$ layer with output $\boldsymbol{y}^{\ell+1}$, and $\phi : \mathbb{R} \to \mathbb{R}$ is a non-linear derivable activation function. According to chain-rule, the gradient is given by [22]

$$\nabla \boldsymbol{y}^{\ell+1} = \text{diag}\big(\phi'(\boldsymbol{W}\boldsymbol{y}^{\ell} + \boldsymbol{b})\big)\boldsymbol{W} \nabla \boldsymbol{y}^{\ell}, \tag{10}$$

where $\phi'$ is the derivative of $\phi$, and $\text{diag}(\boldsymbol{z})$ arrange its input vector $\boldsymbol{z} \in \mathbb{R}^m$ on the diagonal of a square matrix $\mathbb{R}^{m \times m}$. In practice, we use the automatic differentiation module of PyTorch [58] to implement the computation of the gradient $\nabla f(\boldsymbol{x}; \boldsymbol{\theta})$ in the forward pass.

# 4 Experiments

**Architecture**. To learn the neural gradient function, we use a simple neural network similar to [6, 46, 22]. It is composed of eight linear layers and a skip connection from the input to the intermediate layer. Except for the last layer, each linear layer contains 512 hidden units and is equipped with a ReLU activation function. The last layer outputs the signed distance for each point. The parameters of the linear layer are initialized using the geometric initialization [6]. After optimization, we use the network to derive pointwise gradients from the data $P$ (see Fig. 2).

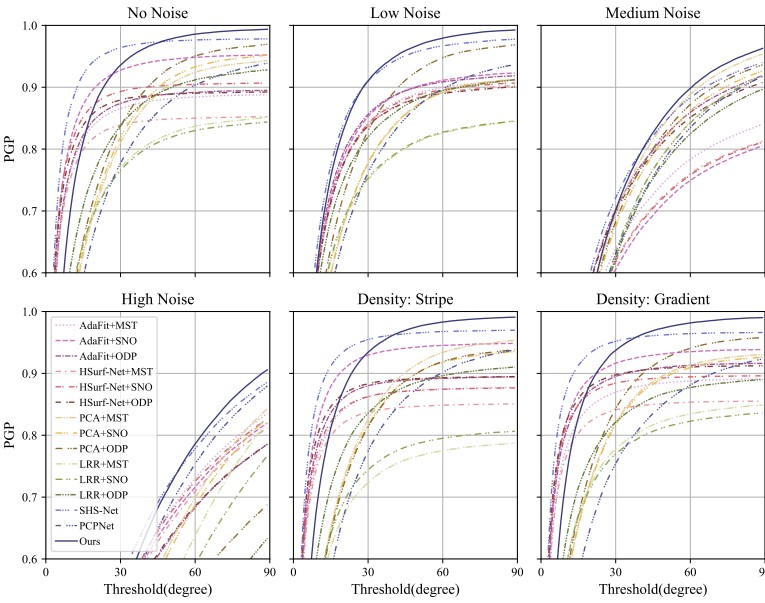

**Distribution** $\mathcal{D}$. We make the distribution $\mathcal{D}$ concentrate in the neighborhood of $P$ in 3D space. Specifically, it is set by uniform sampling points $p$ from $P$ and placing an isotropic Gaussian $N(p, \sigma^2)$ centered at each $p$, with standard deviation parameter $\sigma$ is adaptively set to the distance from the $l$-th nearest point to $p$ [6, 7]. We set $l$ to 25 for the PCPNet dataset and 50 for the FamousShape dataset.

**Implementation**. We choose the neighborhood scale set $\{K_s\}_{s=1}^{N_K}$ as $\{1, K/2, K\}, K = 8$, and set the parameter $\rho$ of adaptive weight to 60. We select $N_Q = 5000$

Figure 5: The PGP curves of oriented normal on the FamousShape dataset. It is plotted by the percentage of good point normals (PGP) for a given angle threshold. Our method has the best value at large thresholds.

points from the distribution $\mathcal{D}$ and $N_G = 2500$ points from $P$ to form the input $\{Q, G\}$. The number of steps is set to $\mathcal{I} = 2$ for all evaluations. As for metric, we use normal angle Root Mean Squared Error (RMSE) to evaluate the estimated normals and Percentage of Good Points (PGP) to show the normal error distribution [40, 39].

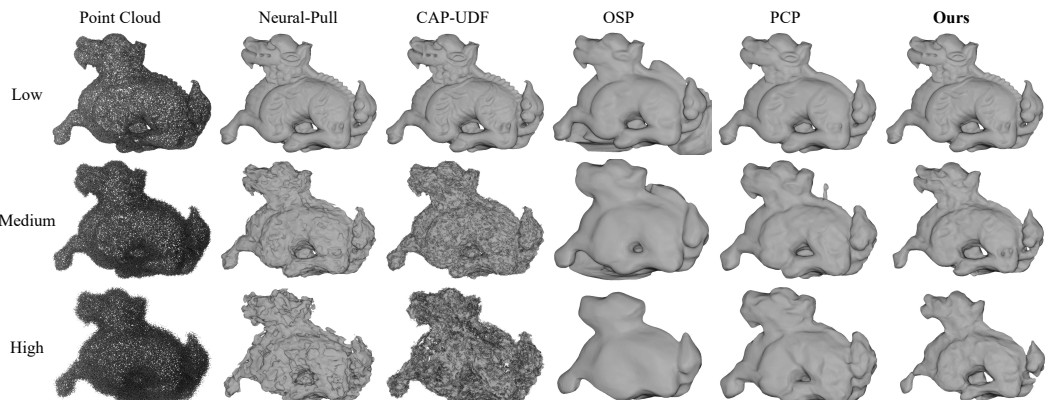

Figure 6: Comparison with surface reconstruction methods using point movement strategy, *e.g.*, Neural-Pull [46], CAP-UDF [85], OSP [47] and PCP [48]. The input point clouds have different levels of noise (low, medium and high).

## 4.1 Performance Evaluation

**Evaluation of Oriented Normal**. In this evaluation, we not only compare with one-stage baseline methods, including PCPNet [24], DPGO [73] and SHS-Net [39], but also compare with two-stage baseline methods that combine representative algorithms based on different design philosophies. Specifically, we use the various combinations of unoriented normal estimation methods (PCA [26], AdaFit [86] and HSurf-Net [40]) and normal orientation methods (MST [26], SNO [67] and ODP [53]). Note that PCPNet, DPGO, SHS-Net, AdaFit and HSurf-Net rely on supervised training with ground truth normals. We report quantitative evaluation results on datasets PCPNet and FamousShape [39] in Table 1. Our method has better performance under the vast majority of data categories (noise levels and density variations), and achieves the best average results. It is worth

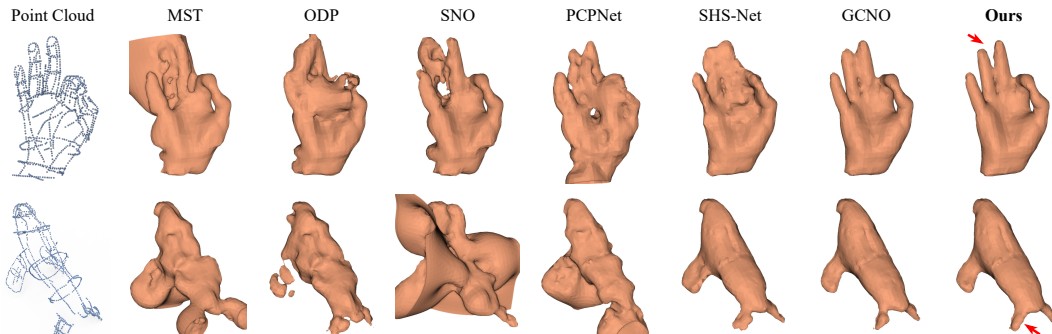

Figure 7: The reconstructed surfaces on wireframe point clouds with sparse and non-uniform sampling. There are less than 1000 points in each input point cloud. The initial normals of MST, ODP and SNO are provided by PCA. GCNO [77] performs similarly to our method.

mentioning that our method achieves a large performance improvement on the FamousShape dataset, which contains different shapes with more complex geometries than the PCPNet dataset. We provide a visual comparison of the estimated normals in Fig. 3, which shows normal vectors of a column section with a gear-like structure. In Fig. 4, we show the normal error map on point clouds with uneven sampling and multi-branch structure. The normal error distribution is shown in Fig. 5. It is clear that our unsupervised method has advantages over baselines in all categories when the angle threshold is larger than $50°$. The evaluation results demonstrate that our method can consistently provide good oriented normals in the presence of various complex structures and density variations.

**Evaluation of Unoriented Normal**. In general, estimating unoriented normals is easier than estimating oriented normals as it does not need to explore more information to determine the orientation, but only focuses on finding the perpendicular of a local plane or surface according to the input patch. Moreover, existing deep learning-based methods for unoriented normal estimation all rely on supervised training with ground truth normals. In contrast, our method is designed for estimating oriented normal in an unsupervised manner. To evaluate the unoriented normals, we compare with the baselines using our oriented normals and ignore the orientation of normals. In Table 2, we report the quantitative evaluation results of supervised and unsupervised methods on datasets PCPNet and FamousShape. We can see that, compared to unsupervised methods, our method achieves significant performance gains on almost all data categories of these two datasets and has the best average results. The supervised methods learn from ground truth normals, and their supervision provides perfect surface perpendiculars, leading to superior results. Our better results than PCPNet highlight the strong learning ability of our method without labels.

**Surface Reconstruction**. For surface reconstruction from point clouds, we compare our method with the latest methods in two different ways. (1) We determine the zero level set of the learned implicit function $f$ via signed distances, and use the marching cubes algorithm [42] to extract the global surface representation of the point cloud. A visual comparison of the extracted surfaces on point clouds with different noise levels is shown in Fig. 6, where our surfaces are much cleaner and more complete under the interference of noise. (2) Based on the estimated oriented normals, we employ

Table 3: Comparison of oriented normal estimation. We use implicit representation methods to estimate oriented normals.

|  | Noise | Density | Average |
|---|---|---|---|
| IGR [22] | 54.77 | 75.90 | 65.33 |
| SAP [60] | 57.56 | 41.32 | 49.44 |
| SAL [6] | 46.69 | 43.78 | 45.24 |
| Neural-Pull [46] | 48.48 | 26.22 | 37.35 |
| Ours | **36.92** | **26.08** | **31.50** |

the Poisson reconstruction algorithm [33] to generate surfaces from wireframe point clouds, and the results in Fig. 7 show that our method can handle extremely sparse and uneven data. The reconstructed surfaces of unsupervised methods on LiDAR point clouds of the KITTI dataset [20] are shown in Fig. 8, where our undistorted surface exhibits a more realistic road scene of real-world data.

Moreover, we repurpose some implicit representation methods, which are designed for surface reconstruction, to estimate oriented normals from point clouds. The comparison of normal RMSE on some selected point clouds in the datasets PCPNet and FamousShape is reported in Table 3. A visual comparison of the reconstructed surfaces on noisy point clouds is shown in the supplementary material. We can see that our method has clear advantages, especially on noisy point clouds. We provide more evaluation results on different data in the supplementary material.

Table 4: Ablation studies for unoriented and oriented normal estimation on the FamousShape dataset.

| Category | | Unoriented Normal — Noise: None | 0.12% | 0.6% | 1.2% | Density: Stripe | Gradient | Average | Oriented Normal — Noise: None | 0.12% | 0.6% | 1.2% | Density: Stripe | Gradient | Average |
|---|---|---|---|---|---|---|---|---|---|---|---|---|---|---|---|
| (a) | w/o $\mathcal{L}_{reg}$ | 13.76 | 16.45 | 31.32 | 42.56 | 13.72 | 12.87 | 21.78 | 16.63 | 19.15 | 36.77 | 55.22 | 16.89 | 16.35 | 26.84 |
| | w/o $\mathcal{L}_d$ | 13.98 | 16.66 | 30.88 | 40.70 | 14.01 | 13.30 | 21.59 | 16.74 | 19.30 | 35.72 | 49.98 | 17.21 | 17.39 | 26.06 |
| | w/o $\mathcal{L}_{con}$ | 13.93 | 17.13 | 33.81 | 45.95 | 13.95 | 13.02 | 22.97 | 23.97 | 26.70 | 40.49 | 69.06 | 23.26 | 17.87 | 33.56 |
| | w/o $\mathcal{L}_v$ | 28.41 | 31.34 | 49.04 | 49.65 | 30.60 | 26.47 | 35.92 | 46.97 | 43.00 | 73.64 | 75.61 | 43.10 | 35.33 | 52.94 |
| | w/o $w$ | 13.76 | 16.56 | 31.45 | 41.61 | 14.00 | 12.83 | 21.70 | 16.81 | 19.40 | 37.06 | 52.97 | 18.12 | 16.11 | 26.74 |
| | w/ $\mathcal{L}_v \cdot w$ | 13.44 | 16.14 | 31.35 | 41.80 | 13.71 | 12.90 | 21.56 | 16.20 | 18.99 | 36.66 | 51.47 | 17.61 | 18.60 | 26.59 |
| (b) | w/o $G$ | 13.86 | 16.73 | 31.10 | 40.73 | 14.02 | 13.30 | 21.62 | 16.99 | 21.66 | 36.70 | 50.30 | 17.37 | 20.35 | 27.23 |
| (c) | $\mathcal{I}=1$ | 14.03 | 17.05 | 33.42 | 45.97 | 13.94 | 13.36 | 22.96 | 23.39 | 26.34 | 40.10 | 63.89 | 19.47 | 27.89 | 33.51 |
| | $\mathcal{I}=3$ | 13.94 | 16.61 | 30.63 | 39.82 | 13.88 | 13.06 | 21.32 | 16.91 | 19.69 | 35.65 | 50.08 | 17.64 | 16.46 | 26.07 |
| (d) | $\{1\}$ | 14.11 | 17.25 | 31.62 | 41.36 | 13.72 | 12.85 | 21.82 | 23.87 | 23.14 | 36.47 | 53.32 | 17.82 | 16.35 | 28.50 |
| | $\{K\}$ | 14.31 | 16.67 | 30.43 | 40.12 | 14.45 | 13.37 | 21.56 | 17.26 | 19.36 | 35.34 | 49.13 | 18.33 | 16.88 | 26.05 |
| | $\{1,K\}$ | 13.72 | 16.62 | 30.77 | 40.28 | 13.92 | 13.06 | 21.39 | 16.54 | 19.40 | 35.65 | 51.98 | 17.02 | 16.55 | 26.19 |
| | $K=4$ | 13.60 | 17.34 | 30.90 | 40.43 | 13.70 | 12.56 | 21.42 | 16.66 | 23.60 | 36.17 | 49.44 | 16.86 | 14.72 | 26.24 |
| | $K=16$ | 14.32 | 16.69 | 30.86 | 41.25 | 14.40 | 13.40 | 21.82 | 17.26 | 19.42 | 35.85 | 50.98 | 17.43 | 17.17 | 26.35 |
| | $K=32$ | 15.02 | 17.18 | 30.61 | 41.15 | 15.44 | 14.26 | 22.28 | 17.82 | 19.79 | 35.70 | 50.64 | 19.28 | 18.22 | 26.91 |
| | **Final** | 13.74 | 16.51 | 31.05 | 40.68 | 13.95 | 13.17 | 21.52 | 16.57 | 19.28 | 36.22 | 50.27 | 17.23 | 17.38 | 26.16 |

## 4.2 Ablation Studies

We aim to achieve the best performance on average in both oriented and unoriented normal estimation. We provide the ablation results on the FamousShape dataset in Table 4 (a)-(d), which are discussed as follows.

**(a) Loss**. We do not use one of the four loss terms in Eq. (9) and the weight $w$ in Eq. (8), respectively. We also try incorporating the weight $w$ into $\mathcal{L}_v(\boldsymbol{\theta})$ of Eq. (7). These ablations do not provide better performance in both oriented and unoriented normal estimation. We observe that the performance drops a lot without using $\mathcal{L}_v(\boldsymbol{\theta})$, and that $\mathcal{L}_{con}(\boldsymbol{\theta})$ has a larger impact on performance. Note that $\mathcal{L}_d(\boldsymbol{\theta})$ plays a more important role on the PCPNet dataset than the FamousShape dataset in our experiments.

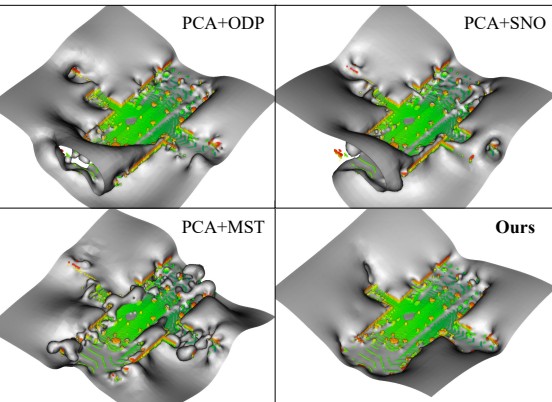

Figure 8: The reconstructed surfaces on the KITTI dataset using estimated normals of unsupervised methods. Points are colored with their heights.

**(b) Input**. We do not sample a subset $G$ from $P$, and take $Q$ as the input instead of $\{Q, G\}$. The results verify the positive effect of $G$ on explicitly indicating the surface and improving performance.

**(c) Iteration**. We set the number of steps $\mathcal{I}$ to 1 and 3, respectively. More iterations bring better performance, but we still choose $\mathcal{I}=2$ after weighing the running time and memory usage.

**(d) Scale Set**. The scale set $\{K_s\}_{s=1}^{N_K}$ in Eq. (6) is chosen to be $\{1, K/2, K\}, K=8$ in our implementation. Here we change the scale set to different sizes, *e.g.*, $\{1\}, \{K\}$ and $\{1, K\}, K=8$, or different values of $K$, *e.g.*, 4, 16 and 32. Some of these settings have advantages in a single evaluation, but do not give better results for both oriented and unoriented normal estimation, such as $\{K\}$ and $\{1, K\}$.

## 5 Conclusion

In this work, we propose to learn neural gradient functions from point clouds to estimate oriented normals without requiring ground truth normals. Specifically, we introduce loss functions to facilitate query points to iteratively reach the moving targets and aggregate onto the approximated surface, thereby learning a global surface representation of the data. Meanwhile, we incorporate gradients into the surface approximation to measure the minimum signed deviation of queries, resulting in a consistent gradient field associated with the surface. Extensive evaluation and ablation experiments are provided, and our excellent results in both unoriented and oriented normal estimation demonstrate the effectiveness of innovations in our design. Future work includes adding more local constraints to further improve the accuracy of normals.

# 6 Acknowledgement

This work was supported by National Key R&D Program of China (2022YFC3800600), the National Natural Science Foundation of China (62272263, 62072268), and in part by Tsinghua-Kuaishou Institute of Future Media Data.

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
