# NeuralGF: Unsupervised Point Normal Estimation by Learning Neural Gradient Function
# — Supplementary Material —

Qing Li[1]    Huifang Feng[2]    Kanle Shi[3]    Yue Gao[1]    Yi Fang[4]
Yu-Shen Liu[1*]    Zhizhong Han[5]

[1]School of Software, Tsinghua University, Beijing, China
[2]School of Informatics, Xiamen University, Xiamen, China
[3]Kuaishou Technology, Beijing, China
[4]Center for Artificial Intelligence and Robotics, New York University Abu Dhabi, Abu Dhabi, UAE
[5]Department of Computer Science, Wayne State University, Detroit, USA
{leoqli, gaoyue, liuyushen}@tsinghua.edu.cn   fenghuifang@stu.xmu.edu.cn
yfang@nyu.edu   h312h@wayne.edu

## 1   Evaluation Metrics

As in [7, 13, 12], we use the Root Mean Squared Error (RMSE) to evaluate the unoriented and oriented normal results of different methods. Specifically, the normal RMSE is calculated as the vector angles between the ground-truth normals $\hat{\mathbf{n}}_i$ and the estimated normals $\mathbf{n}_i$, that is,

$$\text{RMSE}_U = \sqrt{\frac{1}{I}\sum_{i=1}^{I}\big(\arccos(|\hat{\mathbf{n}}_i \odot \mathbf{n}_i|)\big)^2}\ , \tag{1}$$

$$\text{RMSE}_O = \sqrt{\frac{1}{I}\sum_{i=1}^{I}\big(\arccos(\hat{\mathbf{n}}_i \odot \mathbf{n}_i)\big)^2}\ , \tag{2}$$

where $\text{RMSE}_U$ and $\text{RMSE}_O$ are metrics for evaluating unoriented and oriented normal results, respectively. The angular error range in unoriented normal evaluation is between $0°$ and $90°$, while the angular error range in oriented normal evaluation is between $0°$ and $180°$. $I$ is the number of point normals to be evaluated. $\odot$ indicates the inner product of two normal vectors and $|\cdot|$ denotes the absolute value.

In addition, to analyze the error distribution of the normal results, we also use the Percentage of Good Points (PGP) [27, 13, 12], which describes the percentage of good points whose normal angle errors are smaller than several specific angle thresholds. Specifically, the PGP with respect to a threshold $\tau$ is computed by

$$\text{PGP}_U(\tau) = \frac{1}{I}\sum_{i=1}^{I}\mathcal{H}\big(\arccos(|\hat{\mathbf{n}}_i \odot \mathbf{n}_i|) < \tau\big), \tag{3}$$

$$\text{PGP}_O(\tau) = \frac{1}{I}\sum_{i=1}^{I}\mathcal{H}\big(\arccos(\hat{\mathbf{n}}_i \odot \mathbf{n}_i) < \tau\big), \tag{4}$$

where $\text{PGP}_U(\tau)$ and $\text{PGP}_O(\tau)$ are metrics for evaluating unoriented and oriented normal results, respectively. $\mathcal{H}$ represents an indicator function that measures whether the normal angle error is smaller than the given threshold $\tau$.

---

*Corresponding author

37th Conference on Neural Information Processing Systems (NeurIPS 2023).

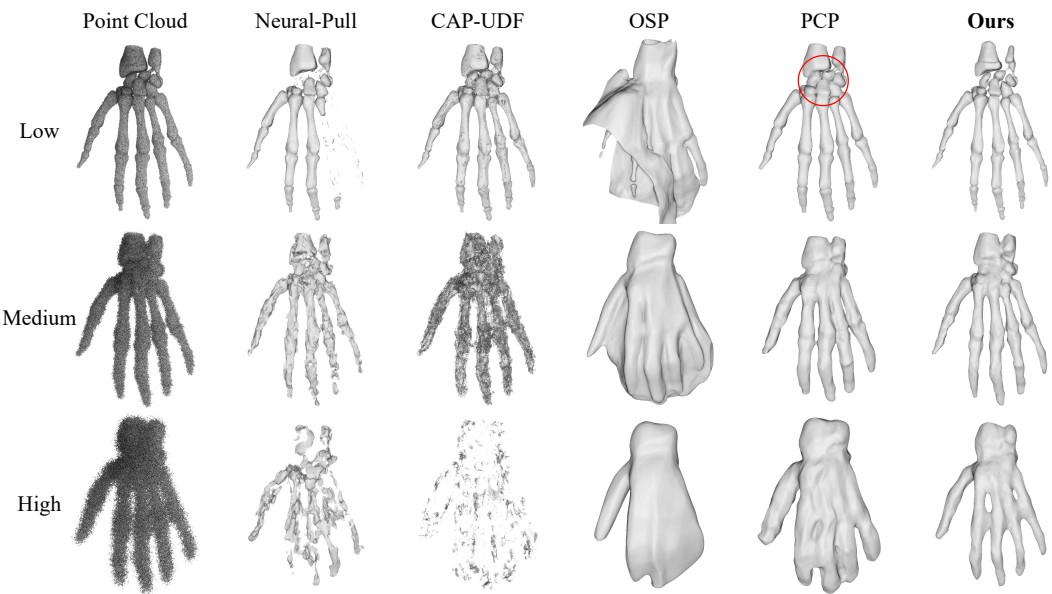

Figure 1: Comparison with surface reconstruction methods using point movement strategy, including Neural-Pull [17], CAP-UDF [26] OSP [18] and PCP [19]. The input point clouds have different levels of noise (low, medium and high).

Table 1: The average normal angle RMSE, number of learnable network parameters, and running time of learning-based oriented normal estimation methods on the PCPNet dataset. We provide optimization time (*i.e.*, training time in the bracket) and inference time of our method. We only provide inference times for other learning-based methods since we directly use their pre-trained network models.

| | PCPNet [7] | HSurf-Net [13] +ODP [20] | AdaFit [27] +ODP [20] | SHS-Net [12] | Ours |
|---|---|---|---|---|---|
| RMSE | 37.66 | 31.07 | 30.93 | 19.79 | **18.70** |
| Time (seconds per 100k points) | 63.02 | 72.47+236.35 | 56.23+248.54 | 65.89 | (1257.37)+**0.10** |
| Parameters (million) | 22.36 | 2.16+0.43 | 4.87+0.43 | 3.27 | **0.46** |
| Relative parameter | **49×** | **6×** | **12×** | **7×** | **1×** |

## 2   Baseline Methods

In the evaluation of unoriented normal estimation, the baseline methods include three types: (1) The traditional normal estimation methods, such as PCA [8], Jet [4] and LRR [25]; (2) The learning-based surface fitting methods, such as DeepFit [2], AdaFit [27] and GraphFit [11]; (3) The learning-based normal regression methods, such as PCPNet [7], Nesti-Net [3], NeAF [14], HSurf-Net [13] and SHS-Net [12]. In the evaluation of oriented normal estimation, the baseline methods include two types: (1) The two-stage based methods that are built by combining three unoriented normal estimation methods (PCA [8], AdaFit [27] and HSurf-Net [13]) and three normal orientation methods (MST [8], ODP [20] and SNO [22]), such as PCA+MST, AdaFit+SNO and HSurf-Net+ODP. (2) The end-to-end based oriented normal estimation methods, such as PCPNet [7] and SHS-Net [12].

## 3   Complexity and Efficiency

In oriented normal evaluation, SHS-Net, PCPNet, HSurf-Net, AdaFit and ODP are learning-based methods, and the others are traditional methods. In this evaluation experiment, we compare our method with the learning-based methods on a machine equipped with NVIDIA 2080 Ti GPUs. As shown in Table 1, we report the average normal angle RMSE, the number of the learnable network parameters, and the running time of each method for oriented normal estimation on the PCPNet dataset [7]. Our method improves the state-of-the-art results while using much fewer parameters. We

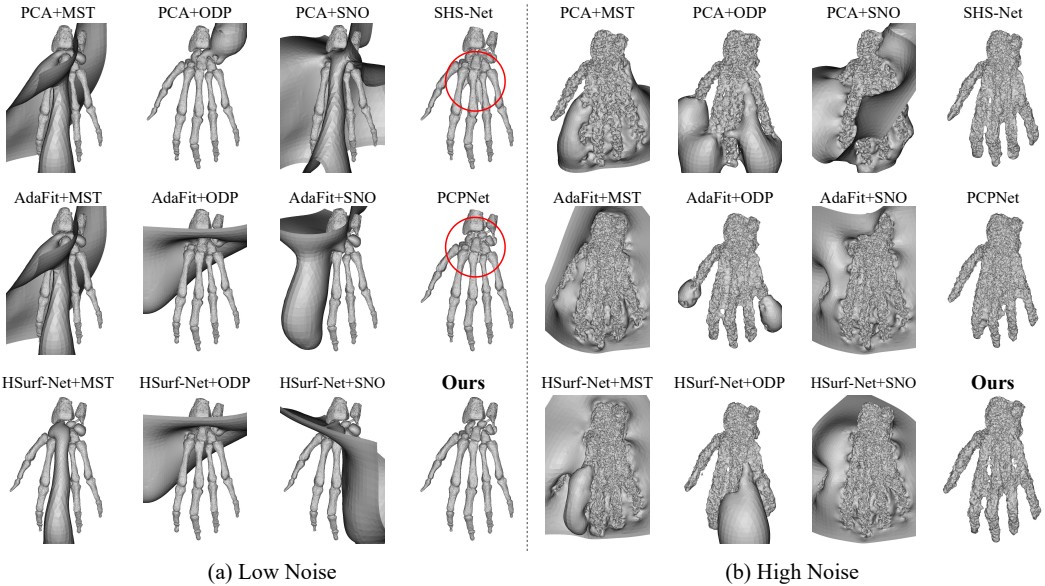

| PCA+MST | PCA+ODP | PCA+SNO | SHS-Net | PCA+MST | PCA+ODP | PCA+SNO | SHS-Net |

| AdaFit+MST | AdaFit+ODP | AdaFit+SNO | PCPNet | AdaFit+MST | AdaFit+ODP | AdaFit+SNO | PCPNet |

| HSurf-Net+MST | HSurf-Net+ODP | HSurf-Net+SNO | **Ours** | HSurf-Net+MST | HSurf-Net+ODP | HSurf-Net+SNO | **Ours** |

(a) Low Noise                  (b) High Noise

Figure 2: Comparison of surface reconstruction results using oriented normals estimated by different methods. The surfaces are reconstructed from point clouds with low noise (a) and high noise (b).

provide the optimization time and inference time of our method, where the optimization time is long and the inference time is very short.

Note that we only report the running time of various methods for predicting normal from point cloud (*i.e.*, testing), and Table 1 does not include the training time of other methods. The other supervised methods can be trained in advance using ground truth, and then their trained models are used for testing. In contrast, our unsupervised method does not require training data and ground truth to train a model, but requires optimization for each shape in the test data to obtain its learned function, so we provide our optimization (*i.e.*, training) time in the table. As for training time, our method has a similar time cost (about 40 hours) to the SOTA method SHS-Net on the entire PCPNet training dataset using an NVIDIA 2080 Ti GPU.

## 4 Surface Reconstruction

We demonstrate the global surface learning and normal estimation capabilities of our method, and verify its reliability by applying the estimated normals to downstream applications, such as surface reconstruction from point clouds. We compare our method with the state-of-the-art methods in two different ways as follows.

**Comparison with Surface Reconstruction Methods**. We determine the zero level set of the learned implicit function $f$ via signed distances, and use the marching cubes algorithm [16] to extract the global surface representation of the point cloud. To evaluate the reconstructed surfaces, we compare our method with baseline methods that are designed for surface reconstruction using point movement strategy, including Neural-Pull [17], CAP-UDF [26], OSP [18] and PCP [19]. A visual comparison of the extracted surfaces on point clouds with different noise levels is shown in Fig. 1. We can see that our reconstructed surfaces have more accurate structures compared to the baseline methods.

**Comparison with Oriented Normal Estimation Methods**. Based on the estimated oriented normals, we employ the Poisson reconstruction algorithm [10] to generate surfaces from point clouds. In Fig 2, we show the reconstructed surfaces on point clouds with different noise levels. In Fig 3, we demonstrate the reconstructed surfaces on point clouds with complex geometries. Compared to baseline methods, the Poisson algorithm can reconstruct more complete geometries with details from various point cloud data based on the accurate normals estimated by our method. In Fig 4, we show the reconstructed surfaces of unsupervised methods on LiDAR point clouds of the KITTI dataset [5], which is captured in outdoor real-world road scenes. The results show that our undistorted surface

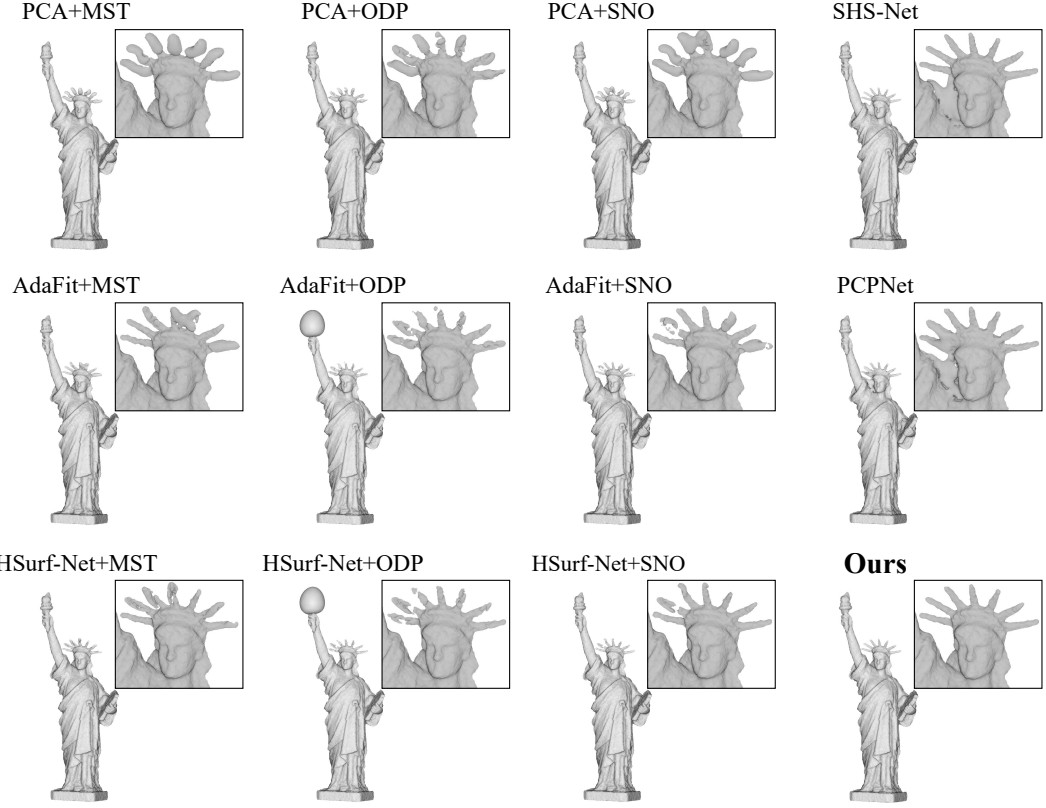

Figure 3: Comparison of surface reconstruction results using oriented normals from different methods. A partially enlarged view is provided for each shape.

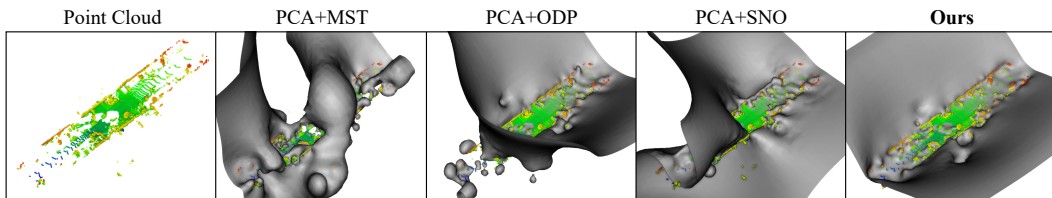

Figure 4: The reconstructed road surfaces on the KITTI dataset using estimated normals of unsupervised methods. Points are colored with their heights.

exhibits a more realistic road scene of real-world data. The comparative results in our experiments demonstrate that our method can accurately estimate oriented normals on point clouds with noise, density variations, and complex geometries, thereby facilitating downstream tasks.

## 5  Limitation, Failure Cases and Broader Impact

Generally, estimating oriented normals with consistent orientations is more challenging than estimating unoriented normals, *i.e.*, finding the perpendiculars of local planes, which has been well-studied over the past decades. The most recent approaches, such as AdaFit and HSurf-Net, can learn to estimate accurate unoriented normal results from point clouds in a supervised manner, but their global orientation consistency can only be achieved by employing a post-processing step, such as MST-based orientation propagation [8]. As we observed in the evaluation experiments, the bottleneck of estimating high-precision oriented normals is correctly determining the normal orientation. Even if the corresponding unoriented normal is very accurate, it is not guaranteed to get a good oriented normal.

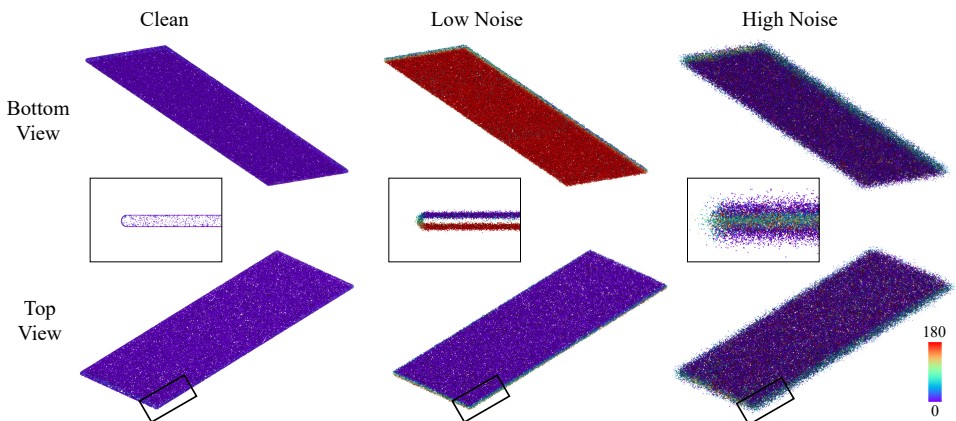

Figure 5: Visualization of oriented normal errors on point clouds with sheet structures, which are taken from the PCPNet dataset [7]. Our method can accurately estimate oriented normals on the clean point cloud, but fail on the point clouds with noise. The unoriented normals on the low noise point cloud are still accurate (shown in red and purple), while the high noise point cloud has many normals with wrong directions (shown in green and yellow). We map the normal angle error to RGB colors. We provide the bottom view, top view, and locally enlarged view of the shape.

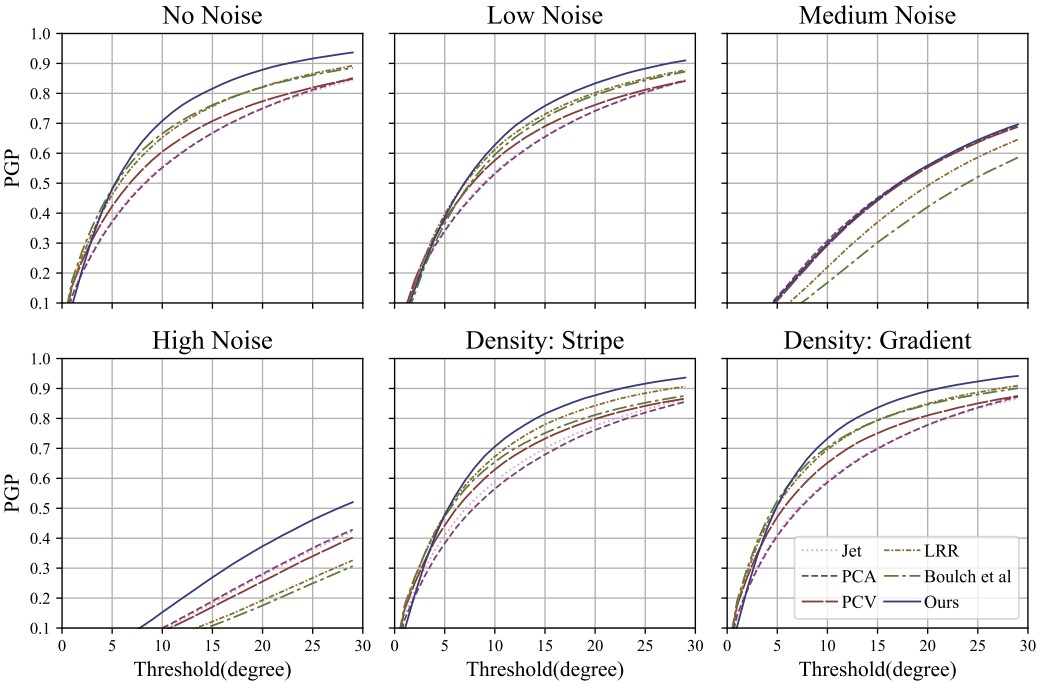

Figure 6: Unoriented normal PGP of unsupervised methods on the FamousShape dataset [12]. The graph is plotted by the percentage of good point normals (PGP) for given angle thresholds. Our method has the best value at large thresholds.

In this work, we propose a novel method to estimate oriented normals directly from 3D point clouds with noise, outliers, and density variations in an unsupervised manner. As we introduced in the paper, our method determines the normal orientation based on learning the neural gradient function, which encourages the neural network to fit global surfaces from the input point clouds and yield consistent unit-norm gradients at each point. In contrast to existing normal estimation methods that use supervised learning of normals to fit surfaces from local patches, the newly proposed approach learns directly from raw data. Our method achieves the state-of-the-art performance while using much fewer network parameters. A limitation of our method is that the neural network needs to fit the model from each single point cloud. The resulting network model cannot be used for shapes not seen during

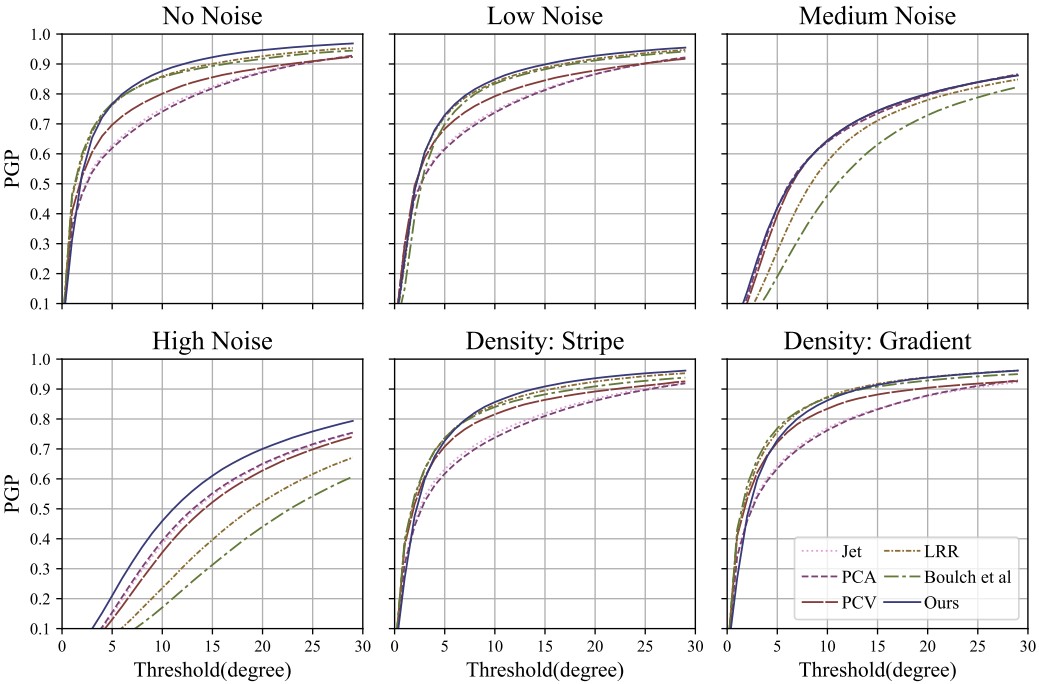

Figure 7: Unoriented normal PGP of unsupervised methods on the PCPNet dataset [7]. The graph is plotted by the percentage of good point normals (PGP) for given angle thresholds. Our method has the best value at large thresholds.

Table 2: Comparison of oriented normal estimation on sparse point clouds.

|  | HSurf-Net [13] +ODP [20] | PCPNet [7] | PCA+ MST [8] | GCNO [24] | SHS-Net [12] | Ours |
|---|---|---|---|---|---|---|
| RMSE | 62.51 | 48.48 | 45.40 | 41.24 | 32.64 | **24.35** |

optimization, and this limitation also exists in some methods of learning implicit functions [17, 26, 19]. Another limitation comes from the construction of query point set $Q$, which is sampled from raw data $P$ according to a distribution $\mathcal{D}$. The default distribution $\mathcal{D}$ in our experiments is suitable for most shapes with various structures. However, we observed that some special structures, such as multi-layer sheets, lead to the inability to estimate oriented normals with consistent orientations. As shown in Fig. 5, we provide some typical cases of these structures. The results shown in the figure are obtained on a clean point cloud and two noisy point clouds. Our method can accurately estimate unoriented normals from these point clouds due to the simple geometry of these shapes. Furthermore, we can see that our method estimates accurate oriented normals on the clean point cloud, but fails on the two noisy point clouds. As shown in the partially enlarged image in the figure, in the clean point cloud, there are clear gaps between the two sheets. In the two noisy point clouds, the points of the two sheets are very close, and even enter into each other's point sets, so that the gap between the two sheets is filled and disappears. This fusion of points will be exacerbated when sampling $Q$ from $P$ using the distribution $\mathcal{D}$, making the model unable to distinguish the inside and the outside of the shape. Eventually, the oriented normals of the two sheets may face the same direction (point cloud with low noise), or even randomly towards the sides of the sheet (point cloud with high noise).

As for the broader impact, various downstream tasks in 3D computer vision can benefit from our method, such as point cloud denoising, surface reconstruction, rendering, segmentation, and so on. The comparison results in Fig. 2-4 demonstrate that our more accurate normals can enhance the performance of the traditional surface reconstruction algorithm. In addition, normal estimation methods are also widely used in the field of robot navigation and grasping, where more reliable normal results can help robots recognize objects and scenes. A potential negative impact is that if the estimated normals are used in some autonomous platforms, it may prevent the controller from making correct decisions when the normal estimator does not provide reliable normals or even fails in some scenarios, resulting in damage or injury.

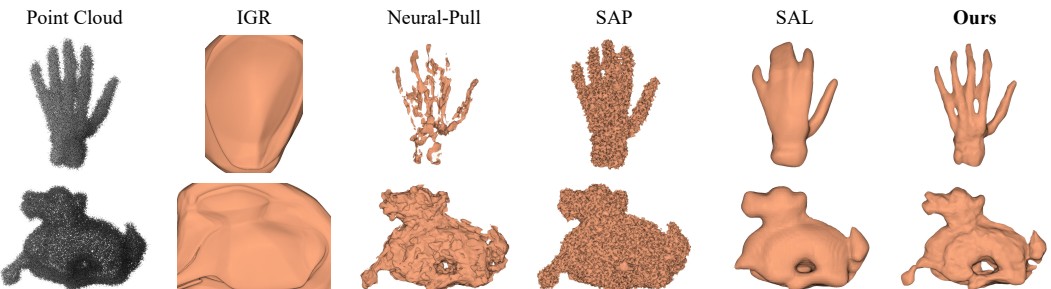

Figure 8: Comparison with implicit representation methods for surface reconstruction, including IGR [6], Neural-Pull [17], SAP [21] and SAL [1]. The input point clouds are noisy.

# 6 More Results

In order to verify the performance on sparse point cloud data, we further conduct an evaluation experiment on a point cloud set that has the same shapes as the FamousShape dataset [12] but each point cloud in this data set has only 5000 points. As shown in Table 2, we provide quantitative comparison results of oriented normal estimation on this sparse point cloud set. It can be observed that our method has the best RMSE result. This evaluation and the comparisons of the challenging cases in the main paper demonstrate the good performance of our method on sparse point clouds.

A visual comparison of the reconstructed surfaces with some implicit representation methods is shown in Fig. 8. In Fig. 6 and Fig. 7, we show the unoriented normal PGP of unsupervised methods on the datasets FamousShape and PCPNet, respectively. The results show that our method has a better performance at the vast majority of thresholds. In Fig. 9, we visualize the position changes of the point cloud after moving through multiple steps in the optimization pipeline. In Fig. 10, we visualize the angle RMSE of the oriented normals estimated by our method on some complicated shapes of the PCPNet dataset [7]. We map the errors to RGB colors to render the point cloud. In Fig. 11- 12, we visualize the angle RMSE of the oriented normals estimated by our method on all shapes of the FamousShape dataset [12].

# 7 More Discussions

## 7.1 Hyperparameter tuning

In all experiments, we use the same network structure and loss weight factors. We did not perform hyperparameter tuning for each shape. We use the same parameters in all categories (clean, noise, and density variation) of a dataset for a fair comparison with other methods. We use the same hyperparameters for all shapes in a benchmark dataset, and only use different parameters to adapt different shapes in different datasets. So, a different input point set does not require different hyperparameters in a dataset and each hyperparameter can be used for a wide range of shapes. Specifically, our method has a hyperparameter needed to be tuned, *i.e.*, determining the standard deviation of distribution $\mathcal{D}$ for different datasets. This hyperparameter is predefined in the code we provide, and we tend to choose a larger value for the dataset with sparse sampling or high curvature.

The hyper-parameter is first set empirically and then tuned according to the experimental results over the validation dataset. The metric we use to tune the hyper-parameter is the RMSE of the oriented normal. Same as existing methods, this metric is also used in evaluation experiments. Specifically, both the PCPNet dataset and the FamousShape dataset contain six categories, we simply choose the average RMSE over the validation dataset as the main indicator when tuning the hyper-parameter for each dataset. As with existing methods, we use standard data splits (training/validation/testing sets) for the datasets used, and the KITTI dataset is only used as the testing set.

## 7.2 Explanation of density-based experiments

The 'gradient' simulates data collected using a 3D scanner, where nearby points are dense while far points are sparse. To achieve this, we give higher weight to points that are closer to the simulated scanner in the probability-based sampling process. The 'stripe' simulates the occlusion situation

Step 0  Step 1  Step 2

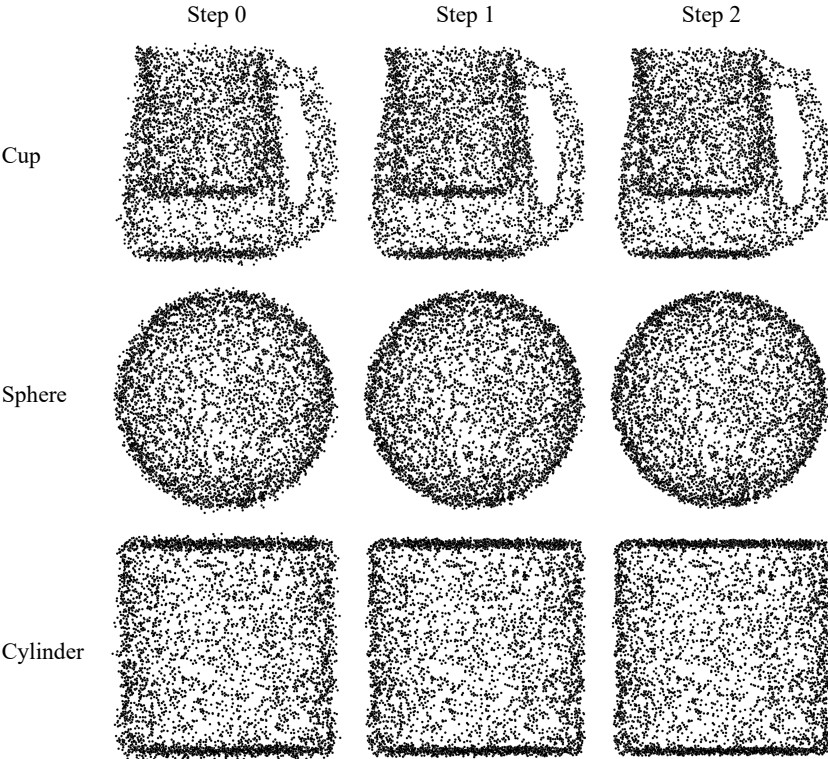

Cup

Sphere

Cylinder

Figure 9: Visualization of the point cloud during the optimization. Our optimization pipeline is formulated as an iterative moving operation of the query point positions. The figure shows the point clouds after moving through multiple steps. During the movement of the second step, the position of the point changes very little. Note that the original input point clouds contain noise and they come from the PCPNet dataset.

during the data collection, making the points in the occluded area sparse or disappear. To achieve that, we divide the shape into multiple areas and sample the points in specific areas with extremely low weights.

### 7.3 Effect of density variation

The optimization of our method is formulated as an iterative moving operation of the input point position. This strategy and the constraints for gradient uniformity of multi-scale neighborhood size make the model have the ability to handle noisy data. During optimization, the input point set $Q$ is generated from the raw point cloud through a probability distribution $\mathcal{D}$, which is built based on the neighborhood of the query points. Therefore, the density variation will affect the input data, and the generated points in sparse areas may be far away from the surface, increasing the difficulty of optimization in the point moving operation. We use the same parameters in all categories (clean, noise, and density variation) of a dataset for a fair comparison with other methods. A solution may be to choose different distributions for clean, noisy and uneven point clouds of a dataset, respectively.

### 7.4 How to obtain the correct normal orientation

The normal orientation is achieved by the gradient of the learned implicit function, and our proposed neural gradient function learns an implicit global surface representation from data. The implicit representation approaches, such as signed distance fields (SDF), represent the surface as zero level set of an implicit function $f$, *i.e.*, $f(x) = 0$. Therefore, we can train a neural network to regress singed distances, where SDF < 0 for inside, SDF > 0 for outside, and SDF = 0 for surface, so that the SDF increases from inside to outside of the surface. Then, the gradient vector field of the SDF is obtained, and the gradient on the iso-surface should have a uniform orientation.

In our method, the optimization is formulated as an iterative moving operation of points to the target surface. According to Neural-Pull [17], the gradient indicates the direction in 3D space in which the signed distance from the surface increases the fastest, so moving a point along or against (decided by SDF) the gradient will find its nearest path to the surface. We can obtain the gradient at each point using the learned SDF, and the gradient is perpendicular to the surface and points to inside or outside based on the initialization of the network.

### 7.5 Clarification of some figures

The output signs in Figure 3 and Figure 8 are not ignored, and the estimated normals are oriented and have consistent orientations. We know that implicit functions can reconstruct artifact surfaces from point clouds with open surface structures, but we do not care about the entire reconstructed surface, and only focus on the regions of existing points on the surface where the SDF can be correctly defined and its gradient has a consistent orientation. For a region without points, whose SDF is uncertain and the zero iso-surface is indeterminate, we do not use the SDF of this region to solve for the gradient. The example in Figure 3 is a part of a full shape with a closed surface, its inside/outside is defined, and we use a section of it for visualization. The example in Figure 8 is a point cloud of the KITTI dataset, the implicit function will learn a closed surface from it, and the points on the surface have a consistent gradient. We only use the SDF at points to solve for gradients as the normals.

### 7.6 Discussion of contributions and improvements to previous works

As we can see from Table 2 of the paper, existing supervised methods can achieve high-precision unoriented normal. However, from Table 1 of the paper, we can observe that higher-precision unoriented normal do not result in more accurate oriented normal using a normal orientation algorithm based on propagation strategies, such as PCA+MST *vs*. AdaFit+MST and PCA+SNO *vs*. HSurf-Net+SNO. This means that even if we develop better unoriented normal estimation algorithms, utilizing existing normal orientation algorithms will not necessarily lead to better orientation results. In brief, the bottleneck of oriented normal estimation is correctly determining the orientation. Moreover, the supervised methods require expensive ground truth as supervision and have been extensively studied, while unsupervised learning of normals is still an unexplored field. Based on the above two observations, we focus on how to use an unsupervised manner to directly learn oriented normals with higher orientation accuracy, instead of learning unoriented normals.

In this work, we introduce a neural gradient function, consisting of the multi-step moving strategy along the gradient in Eq.(2), the gradient uniformity of multi-scale neighborhood size in Eq.(7), and the gradient consistency of multi-step moving in Eq.(8). From the ablation studies, we can see that the performance of the algorithm drops a lot if we do not use these strategies, especially the losses in Eq.(7) and Eq.(8). In Fig.6 of the paper and Fig.1 of the supplementary material, thanks to the losses in Eq.(2) and Eq.(7), our method is more robust against noise compared to surface reconstruction methods.

Existing learning-based normal estimation methods, such as PCPNet, require ground truth normals as supervision for training, and there is still a lot of room for improvement in performance, especially in some challenging cases. Although PCPNet can predict point normal in a forward process with pre-trained parameters, it does not generalize well to unseen cases. We resolve this issue using a per-shape overfitting strategy, which significantly improves the performance in unseen cases. Our method directly learns normals from raw data without using ground truth and achieves better performance for various inputs. In addition, our method can also provide better surface reconstruction results. We do not intend to replace methods, such as PCPNet, that can directly use pre-trained models on different data, but rather as a new exploration that can provide another option for the 3D computer vision community to easily obtain more accurate normals and surfaces from point clouds.

### 7.7 The connection between surface reconstruction methods and normal estimation

In recent years, researchers have paid more attention to the global consistency of normal orientations, such as iterative Poisson Surface Reconstruction (iPSR) [9], Parametric Gauss Reconstruction (PGR) [15] and Stochastic Poisson Surface Reconstruction (SPSR) [23]. In addition to traditional approaches, deep neural networks have been applied to gather information for orientation and reconstruction. Some works learn the implicit function directly from the input point cloud and

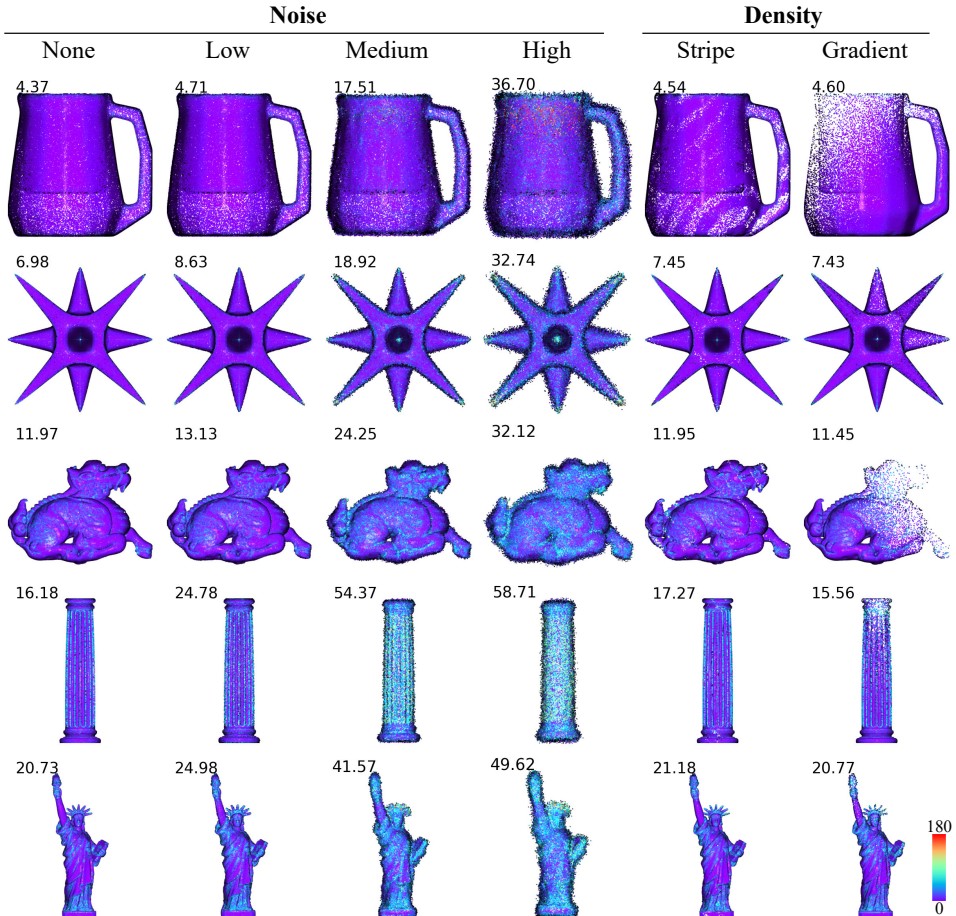

Figure 10: Error visualization of oriented normals estimated by our method on several shapes of the PCPNet dataset [7]. We map the normal errors to a heatmap ranging from 0° (purple) to 180° (red) for visualization. The average RMSE values are reported for each point cloud.

eliminate the need for training data, such as SAL [1], Neural-Pull [17], IGR [6] and SAP [21]. Although these surface reconstruction methods do not aim to estimate point cloud normals, they will add normals (or gradients) to the constraint conditions during the surface optimization process to assist in surface reconstruction. The experimental results of these methods also validate the benefits of using normals in surface reconstruction.

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

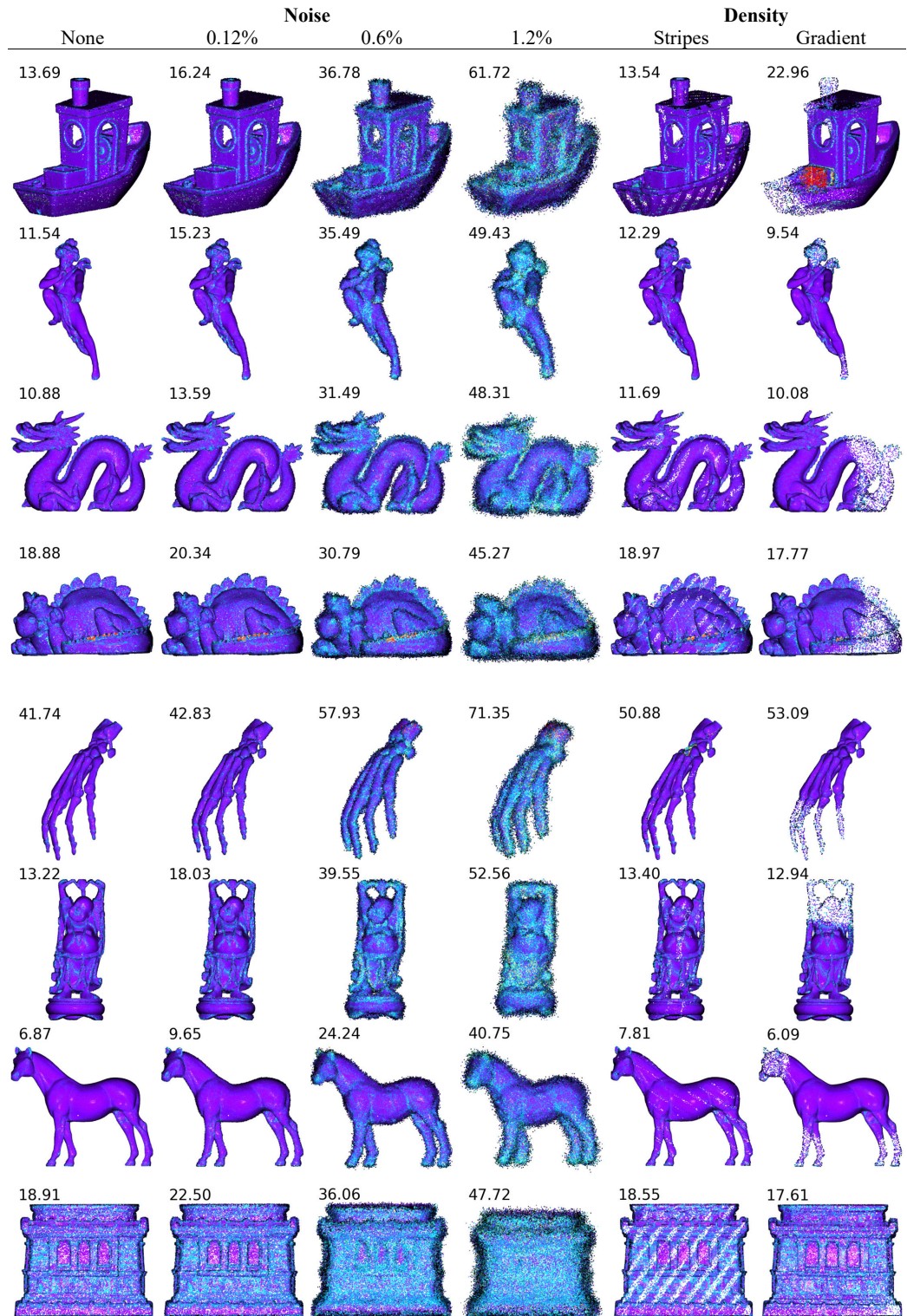

Figure 11: Error visualization of oriented normals estimated by our method on the FamousShape dataset [12]. We map the normal errors to a heatmap ranging from $0°$ (purple) to $180°$ (red) for visualization. The average RMSE values are reported for each point cloud.

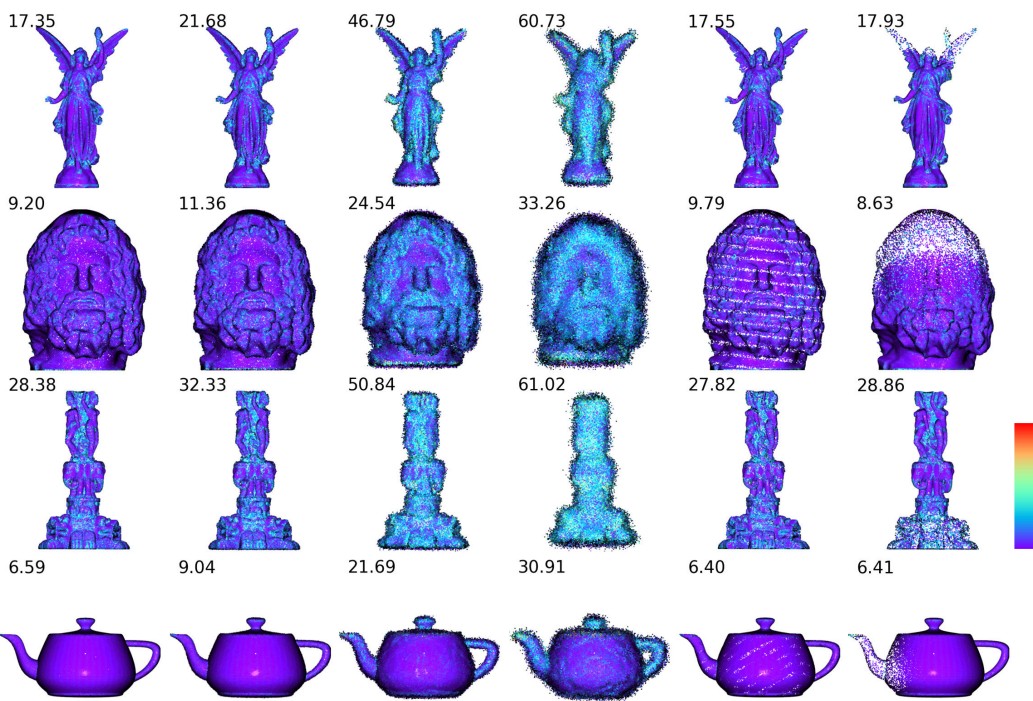

Figure 12: (Same as the previous figure.) Error visualization of oriented normals estimated by our method on the FamousShape dataset [12]. We map the normal errors to a heatmap ranging from $0°$ (purple) to $180°$ (red) for visualization. The average RMSE values are reported for each point cloud.