# OpenReview forum: "NeuralGF: Unsupervised Point Normal Estimation by Learning Neural Gradient Function"
_NeurIPS.cc/2023/Conference — NeurIPS 2023 poster_

### Official Review · Reviewer_6zu3 · 2023-06-19

**Soundness:** 3 good
**Presentation:** 2 fair
**Contribution:** 2 fair
**Rating:** 6
**Confidence:** 3

**Summary:**

The paper proposes a method to learn normals for point clouds using a neural representation. The idea is to perform a multi-step point moving and compute a series of losses to constrain the gradient of the field to describe shape locality with consistency. The paper proposes a large series of comparisons with the existing methods, showing promising results, although the training time is particularly slow.

== POST-REBUTTAL ==

After the discussion, the authors addressed my concerns. From other reviews, I also see a general consensus for acceptance, and the general recommendations are about clarifying some minor aspects. The only negative reviewer seems unwilling to defend the rejection position and does not point to significant weaknesses. Hence, I decided to increase my score and lean toward acceptance. Authors already agreed to incorporate the suggested changes, which are indeed essential; especially, I would stress that the two figures that show partial sections of complete shapes should be clearly described so as to not be deceptive.

I wish authors best of luck with their work.

**Strengths:**

1) The paper obtained good reconstruction results; the shown shapes have varying numbers of different geometries, and the method seems to outperform the competitors.
2) While the method part is not straightforward to understand, the attached code and the implementation details should provide enough details for replicability.

**Weaknesses:**

1) The paper does not convey a clear and precise message about its contribution. I am not an expert in this specific field, and I have quite a hard time understanding what makes this work different from previous ones. Accordingly to the conclusion, the main contribution seems in introducing a new loss/optimization schema. However, from the ablation study (which is also not easy to inspect, given the number of experiments and the lack of a best in bold), the "full" method is not always the clear winner, and it makes unclear the contribution of the losses. From my understanding, the main advancement is in providing multi-scale neighbour consistency, which helps orient all the normals in the same direction. I suggest clearly stating the main insights and adding more structure to 3.2 (e.g., with paragraph titles).

2) Some works seem missing: [A] proposes a smoothness regularization for implicit representations; [B] involves a differentiable Poisson Shape reconstruction. I think a proper discussion would be useful, and especially fostering this kind of discussion in the related works. For example, at the moment, the two paragraph end highlighting that the method is unsupervised, and achieves better performance than the previous works. But how is this obtained? What is the key aspect that enables such advancement and is not considered in the previous works (if any)?



[A]: Implicit Geometric Regularization for Learning Shapes, Implicit Geometric Regularization for Learning Shapes, Gropp et al., ICML 2020
[B]: Shape As Points: A Differentiable Poisson Solver, Peng et al., NeurIPS 2021

**Questions:**

A) From 3.2 I cannot completely understand how the method obtains the correct normal orientation between the two possible directions. Is it obtained thanks to the multi-scale neighbourhood size?
B) From supplementary material, the training time seems dramatically slower than other methods. What is the main cause of that?

**Limitations:**

Limitations are sufficiently discussed in the supplementary material

---

> ### Author Rebuttal · Authors · 2023-08-10
>
> ### 1. Discussion of contributions, and improvements to previous works.
> As we can see from Table 2, existing supervised methods can achieve high-precision unoriented normal.
> However, from Table 1, we can observe that higher-precision unoriented normal do not result in more accurate oriented normal using a normal orientation algorithm based on propagation strategies, such as PCA+MST vs. AdaFit+MST and PCA+QPBO vs. HSurf-Net+QPBO.
> This means that even if we develop better unoriented normal estimation algorithms, utilizing existing normal orientation algorithms will not necessarily lead to better orientation results.
> In brief, the bottleneck of oriented normal estimation is correctly determining the orientation.
> The supervised methods require expensive ground truth as supervision and have been extensively studied, while unsupervised learning of normals is still an unexplored field.
> Based on the above observations, we focus on how to use an unsupervised manner to directly learn oriented normals with higher orientation accuracy, instead of learning unoriented normals.
>
> Both IGR [ICML 2020] and SAP [NeurIPS 2021] learn neural shape representation from point clouds without oriented normal.
> IGR proposes an implicit geometric regularization to favor a smooth zero level set of an implicit function f, i.e., f(x)=0.
> SAP proposes a differentiable poisson solver to represent the shape surface as an oriented point cloud at the zero level set of an indicator function f, i.e., f(x)=0.
> These methods focus on accurately locating the position of the zero iso-surface of f(x) to extract the shape surface.
> However, they ignore the constraints on the gradient of the function during optimizing the neural network.
> We know that the gradient determines the direction of function convergence, and the gradient of the iso-surface can be used as the normal of the surface.
> If the gradient can be guided reasonably, the convergence process will be more robust and efficient, avoiding local extremum caused by noise or outliers.
> Motivated by this, we try to incorporate neural gradients into implicit function learning to achieve oriented normals.
>
> In this work, we introduce a neural gradient function, consisting of the multi-step moving strategy along the gradient in Eq.(2), the gradient uniformity of multi-scale neighborhood size in Eq.(7), and the gradient consistency of multi-step moving in Eq.(8).
> From the ablation studies in Table 3(a), we can see that the performance of the algorithm drops a lot if we do not use these strategies, especially the losses in Eq.(7) and Eq.(8).
> In Fig.6 of the paper and Fig.1 of the supplementary material, thanks to the losses in Eq.(2) and Eq.(7), our method is more robust against noise compared to surface reconstruction methods.
> The multi-scale neighbor consistency mentioned by the reviewer, i.e., Eq.(7), is one of the important advancements of this work.
>
> ### 2. Clarification of ablation study.
> For the ablation studies in Table 3, we report results for both tasks of unoriented and oriented normal estimation.
> The first two categories (a) and (b) in the table are obtained by removing modules (losses and inputs).
> It can be seen that the performance of the algorithm drops relative to the complete method 'Full', especially the losses in Eq.(7) and Eq.(8).
> These studies demonstrate the effectiveness of our novel designs and the contribution of the losses to improving the performance of the algorithm.
> The latter two categories (c) and (d) are experiments on parameters, and they are based on using all losses.
> Some parameter settings provide better results in a single task, but do not give better results for two tasks.
> To handle different cases of more datasets, such as KITTI, we choose the parameters that perform the best on average in both tasks.
> We cannot conclude that the contribution of the losses is unclear because the 'Full' is not always the winner.
>
> ### 3. How to obtain the correct normal orientation.
> The normal orientation is achieved by the gradient of the learned implicit function, and our proposed neural gradient function learns an implicit global surface representation from data.
> The implicit representation approaches, such as signed distance fields (SDF), represent the surface as zero level set of an implicit function f, i.e., f(x)=0.
> Therefore, we can train a neural network to regress singed distances, where SDF<0 for inside, SDF>0 for outside, and SDF=0 for surface, so that the SDF increases from inside to outside of the surface.
> Then, the gradient vector field of the SDF is obtained, and the gradient on the iso-surface should have a uniform orientation.
>
> In our method, the optimization is formulated as an iterative moving operation of points to the target surface.
> According to Neural-Pull [ICML 2021], the gradient indicates the direction in 3D space in which the signed distance from the surface increases the fastest, so moving a point along or against (decided by SDF) the gradient will find its nearest path to the surface.
> We can obtain the gradient at each point using the learned SDF, and the gradient is perpendicular to the surface and points to inside or outside based on the initialization of the network.
>
> ### 4. The training time is slower than other methods.
> Note that we only reported the running time of various methods for predicting normal from point cloud (testing), and Table 1 in supplementary material does not include the training time of other methods.
> The other supervised methods can be trained in advance using ground truth, and then their trained model is used for testing.
> In contrast, our unsupervised method does not require training data and ground truth to train the model, but needs to be optimized based on test data to obtain its learned function, so we provide our optimization (training) time in the table.
> As for training time, our method has a similar time cost (about 40 hours) to the SOTA method SHS-Net on the entire PCPNet dataset using an NVIDIA 2080 Ti GPU.

---

> > ### Comment · Reviewer_6zu3 · 2023-08-14
> > **Post-rebuttal**
> >
> > I thank the authors for their answers to my concerns.
> >
> > I understand that the normal orientation can be recovered using the SDF, which separates the inside from the outside. However, in some of the shown examples, the concept of inside/outside is not well defined (e.g., Figure 3, Figure 7). How are these cases solved? Is the output sign ignored, and the normals are considered unoriented (and visualization is just illustrative)?
> >
> > At the moment, I do not have further questions, and I am considering increasing my score.
> > I am looking forward to reading other reviewers' opinions.

---

> > > ### Author Response · Authors · 2023-08-16
> > >
> > > We thank the reviewer for considering increasing the score.
> > > The output signs in Figure 3 and Figure 7 are not ignored, and the estimated normals are oriented and have consistent orientations.
> > > We know that implicit functions can reconstruct artifact surfaces from point clouds with open surface structures, but we do not care about the entire reconstructed surface, and only focus on the regions of existing points on the surface where the SDF can be correctly defined and its gradient has a consistent orientation.
> > > For a region without points, whose SDF is uncertain and the zero iso-surface is indeterminate, we do not use the SDF of this region to solve for the gradient.
> > > The example in Figure 3 is a part of a full shape with a closed surface, its inside/outside is defined, and we use a section of it for visualization.
> > > The example in Figure 7 is a point cloud of the KITTI dataset, the implicit function will learn a closed surface from it, and the points on the surface have a consistent gradient.
> > > We only use the SDF at points to solve for gradients as the normals.

---

> > > ### Author Response · Authors · 2023-08-21
> > > **We are glad to take more questions**
> > >
> > > Dear reviewer 6zu3,
> > >
> > > We are glad to have your additional comments or take more questions from you. We believe they would be helpful to clarify any ambiguities and increase your rating as you mentioned in the previous comment.
> > >
> > > Thanks,
> > >
> > > Authors

---

> > > ### Comment · Area_Chair_zwft · 2023-08-21
> > >
> > > Dear 6zu3,
> > > after reading the other reviewers' opinions, do you have any final questions for the authors?

---

> > > > ### Comment · Reviewer_6zu3 · 2023-08-21
> > > > **No further questions**
> > > >
> > > > Dear all,
> > > > As mentioned above, I do not have further questions. I do not find main concerns in the occurred discussion, so I increase my score.
> > > >
> > > > Best.

---

> > > > > ### Author Response · Authors · 2023-08-21
> > > > > **Thanks for the rating of acceptance**
> > > > >
> > > > > Dear reviewer 6zu3,
> > > > >
> > > > > Thanks for your valuable feedback and the rating of acceptance. We will revise the paper according to your comments.
> > > > >
> > > > > Best,
> > > > >
> > > > > Authors

---

### Official Review · Reviewer_1E4M · 2023-07-03

**Soundness:** 3 good
**Presentation:** 2 fair
**Contribution:** 3 good
**Rating:** 6
**Confidence:** 4

**Summary:**

The paper introduces a method for estimating oriented normals from a given point cloud by utilizing a neural networkto model a sign distance field (SDF). The proposed approach involves training the SDF representation, which allows for easy querying of gradients at the positions of the point cloud. The method employs a set of loss functions that simulate an iterative process of moving query points to match target points based on the neural gradients. Experiments are conducted on unsupervised oriented normals estimation from input point clouds, which may contain noise, outliers, and density variations.

**Strengths:**

- The authors clearly highlight the limitations of previous works, providing a good motivation for their proposed method. The promising results showcased in Figure 1 further support this motivation.
- The introduction and related work sections are comprehensive, providing the necessary background for understanding the normal estimation task. This makes the paper self-contained.
- The method is presented in a well-structured manner, starting with a high-level overview before delving into the details. This organization helps to keep the reader engaged and informed throughout the paper.

**Weaknesses:**

- The mathematical definitions and notations in the method section need to be revisited. Some crucial definitions, such as $f_i$ in Equation 2, are mentioned without being properly defined, making Sections 3.1 and 3.2 harder to understand. Additionally, the notation $ \{ Q, G\} $ is confusing as the operation involving this set containing two sets of points is not clearly defined or highlighted. The notation $f_i^G$ also requires clarification. Furthermore, the unit gradient $\boldsymbol{n}$ mentioned in Line 164 lacks clear information about its 3D positions.
- The authors refer to the post-processing step of computing the gradients of the learned SDF as "inference". However, "inference" typically refers to the process of applying a trained model on unseen inputs for generalization. This usage of terminology can be misleading and confusing.
- Figure 2 does not fully capture the presented method as it introduces notations that are only defined in the text. It is unclear from the figure alone, even with the caption, what the input to the model is.
- A comparison to other neural SDF methods that learn an SDF from an input point cloud, such as SAL (https://arxiv.org/abs/1911.10414) or IGR (https://arxiv.org/abs/2002.10099), is missing. These methods could be relevant for extracting normals easily, as in the suggested method.
- The paper lacks a reference to SAP (https://pengsongyou.github.io/sap), which provides a scenario of reconstructing normals and a surface from an input point cloud.
- It is recommended to include illustrations for the different losses mentioned, similar to the visualization in Figure 2 (i) at the bottom.

**Questions:**

Please see concerns and questions in weaknesses. Few further questions:

- Are the loss coefficients tuned for different noise levels, or are they fixed throughout the experiments?
- Can the presented method also yield the complete negative solution, i.e., $-\boldsymbol{n}$, which is equivalent up to a global sign? Does this depend on the initialization or other factors?

**Limitations:**

The authors discussed limitation, however the limitation discussion and figure is shown soley in the appendix.

---

> ### Author Rebuttal · Authors · 2023-08-10
>
> ### 1. Some definitions, notations, and wordings need to be revised.
>
> $f_i$ in Eq.(2) represents the signed distance of each point during the i-th position movement process.
> $f_i^G$ in Eq.(3) represents the signed distance of each point in point set G during the i-th position movement process.
>
> As in Line 117, Q and G are two point sets sampled from the raw point cloud in different ways, and they are used as the input of the network during optimization.
>
> The unit gradient n in Line 164 is the normal of each point on the surface.
>
> The wording 'inference' will be changed to 'normal estimation'.
>
> We will revise the paper to further polish the definitions, notations, and wordings.
>
>
> ### 2. Comments on Fig.2, loss and model input.
>
> We will revise Fig.2 to clearly illustrate the method and add an illustration for the designed loss.
>
> During the optimization, the model input is two point sets Q and G sampled from the raw point cloud.
> During the normal estimation, the model input is the entire raw point cloud.
>
>
> ### 3. Comparison with other neural SDF methods.
>
> We will add references to the mentioned methods, namely SAL, IGR and SAP, in the revised version.
>
> We use some implicit representation methods to estimate oriented normals.
> The comparison of normal RMSE on point clouds in the datasets PCPNet and FamousShape is reported in the following table and the visual comparison is shown in *Rebuttal PDF Fig.6*.
> We can see that our method has clear advantages.
>
> We also have compared with some other surface reconstruction methods in Fig.6 of the paper and Fig.1 of the supplementary material.
> Our method can reconstruct better surfaces, especially on noisy point clouds.
> Moreover, our method can handle point clouds with uneven sampling and open surface structure, such as the KITTI dataset in Fig.7.
>
> |Method  |SAP [1]     |IGR [2]     |SAL [3]     |Neural-Pull [4] |Ours       |
> |:-:     |:-:         |:-:         |:-:         |:-:             |:-:        |
> |Noise   |57.56       |54.77       |46.69       |48.48           |***36.92***|
> |Density |41.32       |75.90       |43.78       |26.22           |***26.08***|
> |Average |49.44       |65.33       |45.24       |37.35           |***31.50***|
>
> [1] Peng et al., Shape As Points: A Differentiable Poisson Solver, NeurIPS 2021.
>
> [2] Gropp et al., Implicit Geometric Regularization for Learning Shapes, ICML 2020.
>
> [3] Atzmon et al., SAL: Sign Agnostic Learning of Shapes from Raw Data, CVPR 2020.
>
> [4] Ma et al., Neural-Pull: Learning signed distance functions from point clouds by learning to pull space onto surfaces. ICML 2021.
>
>
> ### 4. More questions.
>
> The coefficients for each loss are fixed across all experiments.
>
> The oriented normal, i.e., gradient, is derived from the learned neural gradient function, and its sign depends on the initialization of the network.
> We do not observe that our method yields the complete negative solution of the sign.
> In the evaluation, if all normals have negative orientations with respect to the ground truth, we can simply reverse their orientation.

---

> > ### Comment · Reviewer_1E4M · 2023-08-19
> > **post rebuttal**
> >
> > Thank you for addressing my main concerns and issues. I encourage the authors to add the mentioned clarifications regarding the notations and improve Figure 2 and the loss illustrations.
> >
> > The additional results presented in the rebuttal are compelling. They should be incorporated in the final submission, especially the comparison to other neural implicit representation methods that struggle to reconstruct from noisy point clouds. As most reviewers mentioned this requirement, it is clear that the paper would benefit from such comparison and discussion in the main paper.
> >
> > Based on these and the other reviews, I raised my rating to week accept. I believe the paper has a solid contribution, but it requires the mentioned amendments in order to be a valid submission.

---

> > > ### Author Response · Authors · 2023-08-21
> > > **Thanks for the final rating of acceptance**
> > >
> > > Dear reviewer 1E4M,
> > >
> > > Thanks for the acceptance rating. We will follow your advice to update our revision.
> > >
> > >
> > > Best,
> > >
> > > Authors

---

> ### Comment · Area_Chair_zwft · 2023-08-17
>
> Dear 1E4M,
> we would love to hear your thoughts. Did the rebuttal and the other reviews change your mind?

---

### Official Review · Reviewer_wJx4 · 2023-07-06

**Soundness:** 3 good
**Presentation:** 3 good
**Contribution:** 3 good
**Rating:** 6
**Confidence:** 4

**Summary:**

In this work the authors present an unsupervised framework for predicting globally consistent, accurate normals given a point cloud as input. The crux of the method is to predict an implicit surface representation (a signed distance function to be exact) from the input point cloud by leveraging the fact that the gradient of the Neural SDF of surface points gives the surface normal at that point. The authors distinguish their method from existing similar approaches by overcoming drawbacks like lack of global consistency by designing a loss function that penalizes incorrect signed predictions and also considers multi-scale neighborhoods. Furthermore, they consider a multi-step iterative approach for refining their surface normal estimations. They show impressive results in terms of RMSE of unoriented/oriented normals across challenging datasets across different levels of sampling densities and noise.

**Strengths:**

1. The proposed method achieves SoTA performance compared to other unsupervised methods on the PCPNet and FamousShape dataset.
2. In the presence of noisy input points, their method is quite competitive even against supervised baselines which is a big plus.
3. Accurately oriented point clouds are hugely sought-after in downstream applications like surface reconstruction and their results on single object surface reconstruction demonstrates the value of their method in this important application.


**Weaknesses:**

1. Comparison with other implicit representations. At its core, the proposed method learns a Neural SDF for the shape represented by the input point cloud and the estimated normals are simply the gradients computed from this Neural SDF using automatic differentiation. As such I believe there should be more comparison drawn to similar methods like Neural-Pull [1], SIREN [2], SAL [3], and Neuralangelo [4]. In particular, [4] uses a finite-difference-based approach to estimate the surface normal during training and I would be curious to see if a similar technique can get similar results in lesser resources for normal estimation.


2. Hyperparameter tuning. Since the method is optimization based, one potential risk is mishandling the hyper-parameter tuning, which turns out to be very important. For example, how does the method choose the hyper-parameter for each shape? Does a different input point set require different hyper-parameters or one hyper-parameter can be used for a wide range of shapes? I think tuning hyper-parameter of a neural field fitting procedure can drastically change the performance, and arguably with proper hyper-parameter tuning, one can probably find a way to curate smooth solutions to different input instances. As a result, it's very essential for this kind of test-time optimization-based method to have a rigorous hyper-parameter tuning procedure reported to eliminate the risk of accidentally adding human judgment into producing the results.

References:

[1] Neural-Pull: Learning Signed Distance Functions from Point Clouds by Learning to Pull Space onto Surfaces. https://arxiv.org/abs/2011.13495
[2] SIREN. https://arxiv.org/abs/2006.09661
[3] SAL. https://arxiv.org/abs/1911.10414
[4] Neuralangelo: High-Fidelity Neural Surface Reconstruction. https://research.nvidia.com/labs/dir/neuralangelo/
[5] NKSR. https://research.nvidia.com/labs/toronto-ai/NKSR/



**Questions:**

1. The density-based experiments are unclear to me. Could the authors please explain what the “gradient” and “stripe” settings exactly imply? The supplementary shows a few qualitative examples but can you explain how they were generated? It also seems that supervised methods perform comparably w.r.t. the proposed method. Could the authors give an intuition for why their method suffers from density variation but not on noisy point clouds (see Table 2 and Fig. 5, especially at low threshold angles)? Which hyperparameters can be tuned to address noise vs. density variation and what is the trade-off for the same? This would help with a better understanding of the authors’ contributions.

2. In the analysis of un-oriented point clouds, we can see that LRR is the second-best performing unsupervised normal estimation method. Yet, we do not see an application of LRR with techniques for introducing orientation like MST in the study on the estimation of oriented normals. Adding LRR + (some orientation method) type of methods to Table 2 and Fig. 5 would make the results a bit more stronger. Basically this can help address the question of why don't we just pick a SoTA unoriented unsupervised point cloud method and do some simple orientation method?


**Limitations:**

Limitation section in the supplementary. Potential other limitations can include shapes with open surface and shapes without volume.

---

> ### Author Rebuttal · Authors · 2023-08-10
>
> ### 1. Comparison with other implicit representations.
>
> We use some implicit representation methods to estimate oriented normals.
> The comparison of normal RMSE on point clouds in the datasets PCPNet and FamousShape is reported in the following table and the visual comparison is shown in *Rebuttal PDF Fig.6*.
> We can see that our method has clear advantages.
>
> We also have compared with some other surface reconstruction methods in Fig.6 of the paper and Fig.1 of the supplementary material.
> Our method can reconstruct better surfaces, especially on noisy point clouds.
> Moreover, our method can handle point clouds with uneven sampling and open surface structure, such as the KITTI dataset in Fig.7.
>
> We will add references to the mentioned methods in the revised version.
> Neuralangelo [5] aims to recover dense 3D surfaces via image-based neural rendering and only provides the source code of a Blender addon.
> Rewriting the algorithm to estimate normals from point clouds is difficult during the short rebuttal period.
>
> |Method  |SAP [1]     |IGR [2]     |SAL [3]     |Neural-Pull [4] |Ours       |
> |:-:     |:-:         |:-:         |:-:         |:-:             |:-:        |
> |Noise   |57.56       |54.77       |46.69       |48.48           |***36.92***|
> |Density |41.32       |75.90       |43.78       |26.22           |***26.08***|
> |Average |49.44       |65.33       |45.24       |37.35           |***31.50***|
>
> [1] Peng et al., Shape As Points: A Differentiable Poisson Solver, NeurIPS 2021.
>
> [2] Gropp et al., Implicit Geometric Regularization for Learning Shapes, ICML 2020.
>
> [3] Atzmon et al., SAL: Sign Agnostic Learning of Shapes from Raw Data, CVPR 2020.
>
> [4] Ma et al., Neural-Pull: Learning signed distance functions from point clouds by learning to pull space onto surfaces. ICML 2021.
>
> [5] Li et al., Neuralangelo: High-Fidelity Neural Surface Reconstruction, CVPR 2023.
>
>
> ### 2. Hyperparameter tuning.
>
> In all experiments, we use the same network structure and loss weight factors.
> We did not perform hyperparameter tuning for each shape.
> We use the same hyperparameters for all shapes in a benchmark dataset, and only use different parameters to adopt different shapes in different datasets.
> So, a different input point set does not require different hyperparameters in a dataset and one hyperparameter can be used for a wide range of shapes.
> Our method has a hyperparameter needed to be tuned, i.e., determining the standard deviation of distribution D for different datasets.
> This hyperparameter is predefined in the code we provide, and we tend to choose a larger value for the dataset with sparse sampling or high curvature.
>
>
> ### 3. Explanation of density-based experiments.
>
> The 'gradient' simulates data collected using a 3D scanner, where nearby points are dense while far points are sparse.
> To achieve this, we give higher weight to points that are closer to the simulated scanner in the probability-based sampling process.
> The 'stripe' simulates the occlusion situation during the data collection, making the points in the occluded area sparse or disappear.
> To achieve that, we divide the shape into multiple areas and sample the points in specific areas with extremely low weights.
>
>
> ### 4. Suffer from density variation at low angle thresholds (Fig.5), but not noise?
>
> The optimization of our method is formulated as an iterative moving operation of the input point position.
> This strategy and the constraints for gradient uniformity of multi-scale neighborhood size make the model can handle noisy data.
> During optimization, the input point set Q is generated from the raw point cloud through a probability distribution D, which is built based on the neighborhood of the query points.
> Therefore, the density variation will affect the input data, and the generated points in sparse areas may be far away from the surface, increasing the difficulty of optimization in the point moving operation.
> As stated in Question 2, we use the same parameters in all categories (clean, noise, and density variation) of a dataset for a fair comparison with other methods.
> A solution may be to choose different distributions for clean, noisy and uneven point clouds of a dataset, respectively.
>
>
> ### 5. Experiment on LRR+(some orientation method).
>
> The RMSE results of LRR+MST/QPBO/ODP on datasets PCPNet and FamousShape are shown in the following table and the curve of PGP is shown in *Rebuttal PDF Fig.4*.
> For oriented normal estimation methods based on two-stage paradigms, the initial unoriented normals will affect the performance of normal orientation.
> From this table, we observe that higher-precision unoriented normals do not result in more accurate oriented normals using a normal orientation algorithm based on propagation strategy, such as PCA+MST vs. LRR+MST.
> This means that even if we use better unoriented normal estimation algorithms (unsupervised or supervised), utilizing existing normal orientation algorithms does not necessarily lead to better orientation results.
>
> |Method      | PCA+MST | PCA+QPBO | PCA+ODP | LRR+MST | LRR+QPBO | LRR+ODP | Ours      |
> |:-:         |:-:      |:-:       |:-:      |:-:      |:-:       |:-:      |:-:        |
> |PCPNet      | 28.52   | 26.52    | 32.16   | 44.82   | 41.98    | 32.44   | **18.70** |
> |FamousShape | 40.48   | 41.31    | 42.92   | 57.83   | 59.84    | 51.93   | **26.16** |

---

> > ### Comment · Reviewer_wJx4 · 2023-08-12
> > **More questions on hyper-parameter tuning**
> >
> > It’s still not very clear to me how a hyper-parameter is chosen for each dataset. Which metric do you use to tune the hyper-parameter? How do you select hold-out set?

---

> > > ### Author Response · Authors · 2023-08-12
> > > **Responses to hyperparameter tuning**
> > >
> > > The hyper-parameter is first set empirically and then tuned according to the experimental results over the validation dataset.
> > > The metric we use to tune the hyper-parameter is the RMSE of the oriented normal.
> > > Same as existing methods, this metric is also used in evaluation experiments.
> > > Specifically, both the PCPNet dataset and the FamousShape dataset contain six categories, we simply choose the average RMSE over the validation dataset as the main indicator when tuning the hyper-parameter for each dataset.
> > > As with existing methods, we use standard data splits (training/validation/testing sets) for the datasets used, and the KITTI dataset is only used as the testing set. We will add more details on hyper-parameter tuning in the revised version.

---

> > > > ### Comment · Reviewer_wJx4 · 2023-08-21
> > > >
> > > > Thanks for the clarification, I'm intending to keep my score.

---

### Official Review · Reviewer_vh4j · 2023-07-07

**Soundness:** 2 fair
**Presentation:** 4 excellent
**Contribution:** 2 fair
**Rating:** 4
**Confidence:** 4

**Summary:**

This work proposes to learn neural gradient functions from point clouds to estimate oriented normal in an unsupervised manner. Specifically, this method introduces several loss functions to constrain query points to iteratively fit the underlying surface, which is defined by the sampled points. Meanwhile, the local gradients are incorporated into the surface approximation to measure the minimum signed deviation of queries, resulting in a consistent normal field associated with the surface. Lastly, some evaluations demonstrate the superior performance of the proposed method over existing approaches.

**Strengths:**

1. This paper is well organized and nicely written. The presentation of motivation is clear and smooth.
2. The work is well motivated. The idea of learning neural gradients for normal estimation is inspiring.
3. Some visual and quantitative evaluation are promising.


**Weaknesses:**

1. I am not sure why we need a distribution D.
2. I do not see any running efficiency statistics, which would be important for a fair assessment.
3. I think there is a missing comparison with [73]. Also, some challenging cases from [73] should be included.


**Questions:**

1. I believe that the designed loss functions can facilitate query points to iteratively reach the moving targets and aggregate onto the approximated surface, thereby learning a global surface representation of the data. However, I am doubting that if the incorporate gradients can achieve a consistent normal field, especially handling some challenging cases. Is it possible to give more details?
2. Both point sets of Q and G are sampled from input point cloud, so in the test stage, how to compute the normal for all points?


**Limitations:**

1. I do not see any failure cases in the paper. It is essential to show some failure cases for the reader to investigate the failure model to benefit future research.

---

> ### Author Rebuttal · Authors · 2023-08-10
>
> ### 1. Why need a distribution D?
>
> The optimization of our method is formulated as an iterative moving operation of the points to the target positions.
> We use this distribution to generate the training data, i.e., query points, which are pulled onto the surface during optimization.
> As in Line 117, the input point set Q is sampled from the raw point cloud via distribution D.
> Line 206 describes how to construct the distribution.
>
>
> ### 2. Running efficiency.
>
> In Section 2 and Table 1 of the **supplementary material**, we have compared the network parameters, and running time of the learning-based methods.
>
>
> ### 3. Comparison with GCNO and using challenging cases of GCNO [73].
>
> We conduct an evaluation on a dataset that has the same shapes as the FamousShape dataset but each shape in this dataset contains only 5000 points.
> As shown in the following table, we report quantitative comparison results of oriented normal estimation on this dataset with sparse point clouds.
> The traditional baseline algorithms, including GCNO and PCA+MST, are implemented in C++ on the Windows platform and run on an Intel i9-11900K CPU.
> Other learning-based methods are implemented in Pytorch on the Linux platform and run on an NVIDIA 2080 Ti GPU.
> We can see that our method has the best RMSE result.
> Comparisons on the challenging cases of GCNO [73] and other cases are shown in ***Rebuttal PDF***.
> These results demonstrate the good performance of our method on sparse point clouds.
>
> |Method |HSurf-Net+ODP |PCA+MST |PCPNet  |SHS-Net  |GCNO   |Ours     |
> |:-:    |:-:           |:-:     |:-:     |:-:      |:-:    |:-:      |
> |RMSE   |62.51         |45.40   |48.48   |32.64    |45.14  |**24.35**|
>
> [73] Xu et al., Globally consistent normal orientation for point clouds by regularizing the winding-number field. ACM TOG 2023.
>
>
> ### 4. Doubt about incorporating gradients to achieve consistent normals.
>
> During the learning of global surface representation, the gradient of the signed distance field determines the convergence direction of the zero iso-surface, and the consistency of the gradient affects the quality of the final result.
> To ensure a continuous and smooth surface, adding constraints to the gradient for optimization can improve the robustness of the algorithm for surface representation and avoid local extremum caused by noise or outliers.
>
> In this work, we introduce the multi-step moving strategy along the gradient in Eq.(2), the gradient uniformity of multi-scale neighborhood size in Eq.(7), and the gradient consistency of multi-step moving in Eq.(8).
> The quantization results in Table 1 and Figure 5, the noisy data in Figure 6, and the uneven data with open structure of the KITTI dataset in Figure 7 indicate that our method can obtain consistent normal under various geometric structures, noise, and non-uniform densities.
> Overall, the shape in the FamousShape dataset has a more complex geometry and topology than the PCPNet dataset, while the point clouds in the KITTI dataset are extremely uneven, sparse, and has an open surface structure.
> The challenging cases provided in Question 3 and more cases in ***Rebuttal PDF*** show the good performance of our method.
> At the same time, we also provided a detailed analysis of the limitation and failure cases of our method in the supplementary material.
>
>
> ### 5. How to compute the normals of all points?
>
> During training, the point sets Q and G sampled from raw point clouds are used as input.
> As in Line 122, during testing, the entire point cloud is fed into the trained network to derive the gradient at each point, and the solved gradient is used as the normal.
>
>
> ### 6. Failure cases.
>
> The failure cases are already provided in Section 4 of the **supplementary material**.
> Our method fails on noisy point clouds of a thin sheet with a hollow structure.
> The integration of noisy points from the upper and lower planes leads to the blurring of the internal and external structures, and the algorithm assumes that all point clouds belong to the same plane.

---

> > ### Author Response · Authors · 2023-08-21
> > **Post Rebuttal**
> >
> > Dear Reviewer vh4j,
> >
> > We have provided a comparison with GCNO [73], used the challenging case of GCNO, and further clarified how incorporating gradients achieves consistent normals. In light of this, we would like to know whether you believe we have addressed your concerns, and if so we hope that you would be willing to increase your score.
> >
> > Thank you for your time,
> >
> > The Authors

---

> > > ### Comment · Area_Chair_zwft · 2023-08-21
> > > **Please provide some feedback regarding rebuttal & other reviews.**
> > >
> > > Dear vh4j,
> > > could you please provide some feedback if the rebuttal addressed your concerns? Do you agree with the other reviewers?
> > > Thanks

---

> ### Comment · Area_Chair_zwft · 2023-08-17
>
> Dear vh4j,
> we would love to hear your thoughts. Did the rebuttal and the other reviews change your mind?

---

### Official Review · Reviewer_HD1H · 2023-07-07

**Soundness:** 3 good
**Presentation:** 3 good
**Contribution:** 2 fair
**Rating:** 6
**Confidence:** 3

**Summary:**

The paper proposes a method that predicts oriented surface normals given a 3D point cloud. The method is based on SDF surface reconstruction. Given a potentially noisy point cloud, a SDF is fitted using an approach similar to Neural-Pull [45]. The surface normal at each point can then be computed based on the gradient of the SDF. Since a global SDF of the shape is recovered, the orientation of the predicted surface normal is also globally consistent. Additionally, the paper proposed various improvements on top of Neural-Pull to improve normal prediction quality: a multi-step movement strategy during optimization, and a multi-scale neighborhood size strategy.

**Strengths:**

* The proposed method is unsupervised -- It is optimized on a per-shape basis and unlike [23,8,39,38], it does not need to be trained on a large collection of shapes, which also avoids any training-evaluation domain gap.
* Unlike local fitting methods [8,82,37,15], the proposed method produces globally consistent surface orientation thanks to the use of a global SDF.
* The proposed method achieved state-of-the-art performance on both oriented and unoriented surface normal estimation. It worked well especially for noisy point clouds.
* The paper has included very comprehensive ablations to show the effects of various design decisions as well as the new components such as multi-step supervision and multi-scale neighbor selection.

**Weaknesses:**

* The proposed method resembles many existing works on point cloud 3D reconstruction, such as SAL [6], Neural-Pull [45], Shape As Points (Peng et al.), NDF (Chibane et al.). In fact, recovering the normal can be considered as the side effect of surface reconstruction -- once the oriented surface is obtained, the oriented normal can be obtained naturally. As there is no comparison on surface normal quality with such methods in the paper, it is not clear if the proposed method has any significant benefit over these methods from  surface reconstruction community.
* Compared to feed-forward methods such as PCPNet [23], the proposed method requires per-shape optimization, which can be computationally expensive.
* The paper will be easier to follow if it can devote some paragraphs to the connections between state-of-the-art surface reconstruction and normal estimation literature.

**Questions:**

* How does the proposed method compare with previous works in terms of speed?
* It is possible to directly repurpose surface reconstruction methods for normal estimation? How would they perform?

**Limitations:**

The limitation and societal impact of the paper is adequately addressed in the supplemental material.

---

> ### Author Rebuttal · Authors · 2023-08-10
>
> ### 1. Comparing with other implicit representation methods and repurposing surface reconstruction methods for normal estimation.
>
> We use some implicit representation methods to estimate oriented normals.
> The comparison of normal RMSE on point clouds in the datasets PCPNet and FamousShape is reported in the following table and the visual comparison is shown in *Rebuttal PDF Fig.6*.
> We can see that our method has clear advantages.
>
> We also have compared with some other surface reconstruction methods in Fig.6 of the paper and Fig.1 of the supplementary material.
> Our method can reconstruct better surfaces, especially on noisy point clouds.
> Moreover, our method can handle point clouds with uneven sampling and open surface structure, such as the KITTI dataset in Fig.7.
>
> |Method  |SAP [1]     |IGR [2]     |SAL [3]     |Neural-Pull [4] |Ours       |
> |:-:     |:-:         |:-:         |:-:         |:-:             |:-:        |
> |Noise   |57.56       |54.77       |46.69       |48.48           |***36.92***|
> |Density |41.32       |75.90       |43.78       |26.22           |***26.08***|
> |Average |49.44       |65.33       |45.24       |37.35           |***31.50***|
>
> [1] Peng et al., Shape As Points: A Differentiable Poisson Solver, NeurIPS 2021.
>
> [2] Gropp et al., Implicit Geometric Regularization for Learning Shapes, ICML 2020.
>
> [3] Atzmon et al., SAL: Sign Agnostic Learning of Shapes from Raw Data, CVPR 2020.
>
> [4] Ma et al., Neural-Pull: Learning signed distance functions from point clouds by learning to pull space onto surfaces. ICML 2021.
>
>
> ### 2. The per-shape optimization is computationally expensive compared to PCPNet.
>
> Existing learning-based normal estimation methods, such as PCPNet, require ground truth normals as supervision for training, and there is still room to improve, especially in some challenging cases.
> Although PCPNet can predict point normal in a forward process with pre-trained parameters, which is much faster than ours, it does not generalize well to unseen cases.
> We resolve this issue using an overfitting strategy, which significantly improves the performance on unseen cases.
> Our method directly learns normals from raw data without using ground truth and achieves better performance for various inputs.
> In addition, our method can also provide better surface reconstruction results.
> We do not intend to replace methods such as PCPNet that can directly use pre-trained models on different data, but rather as a new exploration that can provide another option for the 3D computer vision community to easily obtain more accurate normals and surfaces from point clouds.
>
>
> ### 3. Add a paragraph to the connection between SOTA surface reconstruction and normal estimation.
>
> In recent years, researchers have paid more attention to the global consistency of normal orientations, such as iterative Poisson Surface Reconstruction (iPSR) [Hou et al. 2022], Parametric Gauss Reconstruction (PGR) [Lin et al. 2022] and Stochastic Poisson Surface Reconstruction (SPSR) [Sellán and Jacobson 2022].
> For example, iPSR runs Poisson reconstruction in an iterative manner and updates normals using the generated surface of the last iteration.
> PGR regards normals and surface elements in the Gauss formula as unknown parameters and optimizes the parametric function space.
> In addition to traditional approaches, deep neural networks have been applied to gather information for orientation and reconstruction.
> Some works learn the implicit function directly from the input point cloud and eliminate the need for training data, such as SAL [Atzmon et al. 2020], Neural-Pull [Ma et al. 2021], IGR [Gropp et al. 2020] and SAP [Peng et al. 2021].
> For example, SAP proposes a differentiable Poisson solver to represent shape surfaces as oriented point clouds, and the point positions and normals are updated during the optimization of surface.
> Neural-Pull predicts the signed distance field to move a point along or against the gradient for finding its nearest path to the surface, and its gradient is equivalent to normal.
> Although these surface reconstruction methods do not aim to estimate normals, they will add normals to the constraint conditions during the surface optimization process to assist surface reconstruction.
> The experimental results of these methods also validate the benefits of using normals in surface reconstruction.
>
> We will add more discussions in the revised version.
>
>
> ### 4. Comparison of running speed.
>
> In Section 2 and Table 1 of the **supplementary material**, we have compared the network parameters, and running time of the learning-based methods.

---

> > ### Comment · Reviewer_HD1H · 2023-08-14
> > **Keep my rating**
> >
> > I would like to thank the authors for the response. The rebuttal has solved all my concerns, especially on its advantages over directly using surface reconstruction methods to recover surface normal -- it seems that the proposed method performed significantly better than generic surface reconstruction methods. I would like to retain my rating of weak accept. What prevents me from giving higher rating is mostly due to the exposition. As the proposed method is based on surface reconstruction methods, it will be helpful to compare and contrast the two, instead of trying to describe the proposed method as something new.

---

### Author Rebuttal · Authors · 2023-08-10

We thank the reviewers for their valuable comments, and our responses to all reviewers are as follows. More visual comparison results are shown in the provided PDF.

---

### Decision · Program_Chairs · 2023-09-21

**Decision:**

Accept (poster)

**Comment:**

The revievers appreciated the strong results, clear presentation, and comprehensive ablations.
A major concern shared by reviewers was the relation to SDF-based surface reconstruction methods, which they found insufficiently discussed. Moreover, experimental comparisons to surface-reconstruction-based approaches and on sparse point clouds were missing. The rebuttal addressed these concerns and experimental results confirmed the superior performance of the proposed method over baselines. All reviewers except for unresponsive vh4j found their concerns addressed in the rebuttal and raised their score. The AC agrees with the majority of the reviewers and recommends acceptance.